# Research Progress of Elastomer Materials and Application of Elastomers in Drilling Fluid

**DOI:** 10.3390/polym15040918

**Published:** 2023-02-12

**Authors:** Lili Yang, Zhiting Ou, Guancheng Jiang

**Affiliations:** National Engineering Research Center of Oil and Gas Drilling and Completion Technology, State Key Laboratory of Petroleum Resources and Prospecting, Ministry of Education (MOE) Key Laboratory of Petroleum Engineering, China University of Petroleum, Beijing 102249, China

**Keywords:** elastomer, drilling fluid, elastic graphite, polyurethane elastomer, epoxy elastomer

## Abstract

An elastomer is a material that undergoes large deformation under force and quickly recovers its approximate initial shape and size after withdrawing the external force. Furthermore, an elastomer can heal itself and increase volume when in contact with certain liquids. They have been widely used as sealing elements and packers in different oil drilling and development operations. With the development of drilling fluids, elastomer materials have also been gradually used as drilling fluid additives in drilling engineering practices. According to the material type classification, elastomer materials can be divided into polyurethane elastomer, epoxy elastomer, nanocomposite elastomer, rubber elastomer, etc. According to the function classification, elastomers can be divided into self-healing elastomers, expansion elastomers, etc. This paper systematically introduces the research progress of elastomer materials based on material type classification and functional classification. Combined with the requirements for drilling fluid additives in drilling fluid application practice, the application prospects of elastomer materials in drilling fluid plugging, fluid loss reduction, and lubrication are discussed. Oil-absorbing expansion and water-absorbing expansion elastomer materials, such as polyurethane, can be used as lost circulation materials, and enter the downhole to absorb water or absorb oil to expand, forming an overall high-strength elastomer to plug the leakage channel. When graphene/nano-composite material is used as a fluid loss additive, flexibility and elasticity facilitate the elastomer particles to enter the pores of the filter cake under the action of differential pressure, block a part of the larger pores, and thus, reduce the water loss, while it would not greatly change the rheology of drilling fluid. As a lubricating material, elastic graphite can form a protective film on the borehole wall, smooth the borehole wall, behaving like a scaly film, so that the sliding friction between the metal surface of the drill pipe and the casing becomes the sliding friction between the graphite flakes, thereby reducing the friction of the drilling fluid. Self-healing elastomers can be healed after being damaged by external forces, making drilling fluid technology more intelligent. The research and application of elastomer materials in the field of drilling fluid will promote the ability of drilling fluid to cope with complex formation changes, which is of great significance in the engineering development of oil and gas wells.

## 1. Introduction

Elastomers are elastic polymers, generally divided into two categories: a thermoplastic elastomer (TPE) and a thermosetting elastomer. A thermosetting elastomer is the traditional rubber. TPE shows rubber elasticity at room temperature and can be plasticized at high temperature. Therefore, this type of polymer material combines the characteristics of thermoplastic rubber and thermoplastic plastics. The basic structural feature of a TPE polymer chain is that it is comprised of some plastic segments (hard segments) and rubber segments (soft segments) originating from different chemical compositions. The interaction between the hard segments is sufficient to condense into microdomains (such as glassy or crystalline microdomains) to form physical crosslinks between molecules. While the soft segment is a segment with high rotation ability. Since the 1990s, increasingly more commercial applications of elastomers have emerged.

By regulating the hard and soft segments, a variety of elastomer materials can be obtained, which have great importance for research and applications. This paper primarily introduces the type of thermoplastic elastomer materials, and discusses the research progress of elastomer materials. According to the classification of material types, elastomers can be divided into polyurethane elastomers, epoxy elastomers, nano-composite elastomers, rubber elastomers, etc.; according to the function classification, elastomers can be divided into self-healing elastomers, expansion elastomers, etc. Among them, polyurethane and other oil-absorbing expansion and water-absorbing expansion elastomer materials are used as LCMs for drilling fluids, graphene/nano-composite materials are used as filtrate reducers, and elastic graphite and other materials are used as drilling fluid lubricants, which have been studied by some researches. The potential of other types of elastomer materials as drilling fluid additives has also great potential for further research.

## 2. Research Progress of Elastomer Materials

Traditional elastomers can be roughly divided into two categories according to the different crosslinking methods: one is the chemical crosslinking of macromolecular chains through covalent bonds (such as polysulfide bonds, carbon-carbon bonds and bonds), represented by vulcanized rubber; the other is the physical crosslinking of flexible macromolecular chains through crystalline microdomains (except for hydrogen bonds), represented by thermoplastic elastomers (TPE) [1]. With the progress of research, supramolecular elastomers based on hydrogen bonding [2], viscoelastic polymers [3] and magnetic response liquid crystal elastomers [4] have been produced. This paper reviews the research progress of elastomer materials and their application in drilling fluids, and introduces the application of elastomers in drilling fluids, as shown in Figure 1.

### 2.1. Urethane Elastomer

#### 2.1.1. Thermoplastic Polyurethane Elastomer

Thermoplastic elastomer material was the earliest elastomer material that went into production. It is a blend of rubber, plastic and synthetic resin. It is also called rubber and plastic. Through the mechanical blending of rubber with some plastics or synthetic resins, the modification of rubber and plastic is finally achieved. Based on the development of rubber and plastic blending technology, a new type of thermoplastic elastomer material with both rubber and plastic properties was successfully developed in the early 1970s, which was called blended thermoplastic elastomer. So far, this kind of thermoplastic elastomer has become a polymer elastomer material with good performances, superior functions and wide applications. Thermoplastic elastomer materials are divided into general TPE and engineering plastic TPE. The general TPE includes olefins, diene, styrene and vinyl chloride, and the engineering TPE includes polyurethane, polyester and amide. Among them, polyurethane TPE is a thermoplastic polyurethane rubber composed of urethane-bonded hard segments and polyester or polyether soft segments, referred to as TPU, with excellent mechanical strength, oil resistance and flexibility. The disadvantages are poor thermal resistance, hot water resistance and compression resistance [5].

Dynamic vulcanized thermoplastic elastomer (TPV) is a special kind of thermoplastic elastomer. It is prepared by the dynamic vulcanization of a mixture of thermoplastic resin and elastomer. The so-called dynamic vulcanization is when rubber and resin are blended. By means of the vulcanization reaction process of a vulcanizing agent (crosslinking agent) under strong mechanical shear stress, the crosslinked rubber particles are dispersed in a continuous thermoplastic matrix in a high mass fraction (50~70%) and finely dispersed form (particle size 1–3 μm). Compared with ordinary thermoplastic elastomers, the rubber components are completely vulcanized and uniformly dispersed in the thermoplastic matrix, so that they have good physical and mechanical properties, processing properties, gas resistance, heat resistance and oil resistance. With the continuous development and expansion of TPV applications. Traditional TPV can no longer meet its needs. Therefore, it is necessary to modify TPV in various aspects to obtain high performance and new, functional TPV to expand its research and application range [6].

Thermoplastic polyurethane elastomer (TPU) has become one of the most important thermoplastic elastomer materials due to its excellent performances and wide applications. TPU is a kind of polyurethane that can be plasticized by heating and dissolution by solvent. The main chains are basically linear with no or few chemical crosslinking. There are many hydrogen bonds between the linear polyurethane molecular chains, which are physically crosslinked. Hydrogen bonds strengthen their morphology, thus giving TPU many excellent properties, such as high modulus, high strength, excellent wear resistance, chemical resistance, hydrolysis resistance, high and low-temperature resistance and mildew resistance. These good performances make thermoplastic polyurethane widely used in shoes, cables, clothing, automotive, medical and health, pipes, films and sheets, and many other fields. TPU-based products do not generally require vulcanization crosslinking, which can shorten the reaction cycle and reduce energy consumption. Because of the linear structure, it can be processed with the same technology and equipment similar to thermoplastics, such as injection molding, extrusion, blow molding, calendaring, etc., especially suitable for the mass production of small- and medium-sized parts. TPU waste materials can be recycled and reused, and different additives or fillers can be used during production or processing to improve certain physical properties and reduce costs.

TPU elastomer not only has the rubber properties of high strength and high wear resistance of crosslinked polyurethane, but also has the thermal plasticity of linear polymer materials; consequently, the application can be extended to the field of plastics. Because of its excellent properties of both rubber and plastic, TPU has boomed in many fields; its development speed is very fast, especially in China. By the end of 2011, Chinese TPU production capacity reached 200,000 tons and the demand was 170,000 tons, accounting for 25% of global demand; the average annual growth rate was maintained at more than 10% and this proportion is still increasing. This paper mainly introduces the synthesis process, performance structure, new technology and new applications of TPU.

##### Synthesis Process of TPU

The raw materials of TPU are bifunctional reactants, and the types of commonly used raw materials are shown in Table 1. Oligomeric diols constitute the soft segment structure of the TPU chain segment, accounting for about 50–80% of the molecular weight of TPU, which has a great influence on the physical and chemical properties of TPU. It is required that the structure of oligomer diol is uniform, with no substituent, which is beneficial to the aggregation of oligomer diol and improves the physical properties of TPU. On the contrary, if the chain segment structure is irregular or there are substituents, it will form an amorphous TPU, resulting in TPU soft segments and hard segments entangled with each other, and cannot be effectively separated, reducing the physical properties of TPU. According to the different uses, different oligomer diols can be selected, and their molecular weights are usually 500–4000. The molecular weight of oligomer diol greatly influences the microphase separation and crystal formation of the TPU structure. When the soft segment molecular weight is greater than 2000 or the hard segment content is less than 24%, the soft segment of TPU can also form a crystalline state. For example, with the increase in the molecular weight of polyester, the number of methylene groups in the molecular chain increases, the intermolecular force and hydrogen bond crosslinking increase, the crystallinity of the chain segment increases, the mechanical properties such as hardness and tensile strength of TPU increase and the elongation at break decreases.

The oligomer diol commonly used in TPU is polyester type. The prepared TPU has high strength, excellent wear resistance, solvent resistance and mold resistance, and hydrolysis resistance is moderate. TPU synthesized from poly(tetramethylene) glycol (PTMG) has excellent elasticity and low-temperature resistance [7]; polybutadiene polyol TPU has high water resistance and electrical insulation. TPU synthesized from polycarbonate polyols has high strength, excellent hydrolysis resistance and abrasion resistance, but the cost is high [8]. Poly(ε-caprolactone) diol TPU has good comprehensive performance, but its cost restricts applications [9]. These oligomeric diols can be used as a soft segment of the polymer alone or can be used to synthesize polyurethane, The reaction to form urethane group is shown in Figure 2, thus realizing many properties and satisfying extensive application fields. In PTMG mixed with PPG polyols, on the one hand, the side groups of the soft segment are increased due to the introduction of PPG, increasing the hydrogen donor group in the hard segment and the steric hindrance of polyether oxygen atoms in the soft segment, thereby reducing the formation of hydrogen bonds between hard and soft segments, reducing viscosity while promoting microphase separation, improving the low-temperature resistance of TPU. On the other hand, the existence of methyl branches also hinders the complete separation of soft and hard segments, reduces the orderly arrangement between elastomers and is detrimental to some properties of TPU. Therefore, in the synthesis of TPU, the type and proportion of polyols should be determined according to the specific application environment and performance requirements to achieve the best mechanical properties and processing operation.

Table 2 of the total. The commonly used diisocyanate for TPU is mainly 4,4′-diphenylmethane diisocyanate (MDI). There are two benzene rings in the molecular structure of MDI, and two NCO groups are symmetrically attached to the 4-position of the two benzene rings. This highly symmetrical structure not only gives TPU rigidity, but also causes crystallization of the hard segment. The high order of the hard segment also leads to a microphase separation between the soft and hard segments, thus giving TPU excellent physical and mechanical properties. In addition, compared with TDI, MDI has a larger molecular weight and lower vapor pressure, and the resulting TPU is more environmentally friendly [9].

Commonly used TPU elastomer chain extenders include aliphatic or aromatic molecules diol or diamine; its molecular weight is generally 80~350. Among them, 1,4-T diol (BDO) is mostly used as a diol chain extender [10], and 3,3′-dichloro-4,4′-diamino-diphenylmethane (MOCA) or ethylene diamine are mostly used as diamine chain extenders [11]. However, TPU with an MOCA chain extender has the disadvantage of poor thermal fatigue resistance. Moreover, the MOCA chain extender leads to a too fast reaction rate, which is not conducive to process control and is suspected to be carcinogenic. During the synthesis of TPU, the chain extender reacts with diisocyanate to form a hard segment rich in urethane groups, forming a hydrogen bond aggregation zone. Li Yue et al. [12] synthesized prepolymer with a NCO mass fraction of 10–16% using poly(tetrahydrofuran) ether as a soft segment and reacted with MDI, followed by reacting with different diol chain extenders to prepare the thermoplastic polyurethane. Table 2 shows the effect of different chain extenders on the mechanical properties of TPU. The results show that the TPU synthesized by a short chain extender has better mechanical properties than that synthesized by a long chain extender. This is because the introduction of a short-chain extender shortens the distance between hard segments and makes the hydrogen bonding between hard segments stronger, which is conducive to the aggregation of hard segment molecules and promotes the microphase separation of TPU molecules, thus giving TPU better mechanical properties.

#### 2.1.2. Silicone Polyurethane Elastomer

Silicone polymers are polymers containing silicon elements in the molecular structure, with repeated Si-O bonds as the main chain, and the silicon atoms are connected with organic groups. Poly(siloxane) (such as silicone oil, silicone rubber and silicone resin, etc.) has many unique properties, such as excellent low-temperature resistance, weather resistance, high-temperature resistance, aging resistance and hydrophobic properties, an extremely broad range of applications, and it has played an important role in the national economy [13].

Silicone elastomers have always been of great interest in the field of new materials. Silicone-modified hard and soft segment elastomer usually utilizes poly(siloxane) as a soft segment, graft or block to other hard segments, such as AB, ABA or (AB)_n_ type. Compared with pure poly(siloxane), the mechanical properties of modified poly(siloxane) are enhanced to certain degrees, which can improve the mechanical strength, adhesion and aging resistance compared to original siloxane polymers. This block polymer is usually composed of rubbery soft segments and glassy or crystalline hard segments, showing a two-phase microstructure, in which the hard segments play a role in physical crosslinking and strengthening the structure of the polymer. More importantly, modified silicone polymers with soft and hard segment structures can form reversible elastomer structures that can be reprocessed by melting or dissolution [14]. Modified silicone polymers with soft and hard segment structures or other silicone-modified polymers include silicone-modified polyurea, polyurethane, polyether, polycarbonate, polyimide and polyolefin. A relatively large number of variables of this block silicone elastomer can be independently controlled, such as the skeleton chemical bond structure, segment molecular weight and total molecular weight of the copolymer, to design materials with target properties. Therefore, the silicone-modified hard and soft segment elastomers have flexible modifiability.

For silicone polyurethane elastomer, the poly(siloxane) is used as a soft segment, while polyurethane segment is used as a hard segment. The polarity of a poly(siloxane) segment is relatively small, and the polyurethane group has a strong polarity. Poly(siloxane) makes the copolymer soft and elastic, while a polyurethane segment plays the role of crosslinking the point and filling. The organic combination of soft and hard segments makes silicone-modified polyurethane elastomer materials exhibit better mechanical strength and oil resistance [15].

Sun et al. [16] used different kinds of solubilizers to modify the polarity, symmetry and other properties of the hard segment, thereby changing the solubility parameters and crystallization behavior of the hard segment, improving the compatibility between the poly(siloxane) and the hard segment, reducing the phase separation degree between the soft and hard segments, and improving the performance of the silicone-polyurethane thermoplastic elastomer. The prepared silicone-polyurethane thermoplastic elastomer not only has good hydrophobic properties and low temperature resistance, but also the mechanical properties of silicone-polyurethane thermoplastic elastomers compared with pure poly(siloxane), and can be used as biomedical materials.

Cheng [17] used polypropylene glycol and isophone diisocyanate as raw materials, dilaurate as catalysts to prepare polyurethane acrylate prepolymer and then added hydroxyl-terminated poly(siloxane) as the soft segment and hydroxyethyl acrylate as the capping agent to synthesize UV-curable polyether modified poly(siloxane) polyurethane acrylate prepolymer (PESiUA), which can be used as polyvinyl chloride leather finishing agent. The prepolymer has excellent compatibility with acrylate monomers and good yellowing resistance. The increase in monomer functionality can improve thermal stability, tensile strength and reduce contact angle and elongation at break. More importantly, due to the advantages of the poly(siloxane) segment and polypropylene glycol segment together with photopolymerization technology, the PVC leather finishing agent based on PESiUA has excellent performance and great application potential.

### 2.2. Epoxy Elastomer

#### 2.2.1. Silicone-Modified Polyurea Elastomer

Polyurea materials have been widely used in road and bridge engineering and other fields, but pure polyurea compounds face poor weather resistance in harsh environments. Silicone-modified polyurea materials can overcome the shortcomings of the poor weather resistance of polyurea materials. Block copolymers containing polyurea segments have higher solubility parameters and can produce greater phase separation with poly(siloxane) segments. Therefore, the study of polyurea-poly(siloxane) block copolymer has certain theoretical and practical significance.

Pang et al. prepared poly(siloxane)-polyether (PPO-PDMS-PPO) block copolymer by Aza-Michael addition reaction [18] using acryloyl methyl-terminated PDMS and amino-terminated PPO as raw materials. A novel polyurea-poly(siloxane)-polyether polymer was prepared by pre polymerization using the synthesized PPO-PDMS-PPO copolymer as the soft segment and 4,4-diisocyanate dicyclohexylmethane (HMDI) and 2-methyl-1,5-pentanediamine (DY) as raw materials.

The results of dynamic mechanical analysis showed that the temperature of the rubber platform region of the polymer was between −50 °C and 80 °C, independent of the content of the hard segment and the molecular weight of PDMS. The viscoelasticity of the obtained soft and hard segment polymers was related to the molecular weight of PDMS in the PPO-PDMS-PPO segment. Compared with the traditional poly(siloxane)-polyurea elastomer, this new polyurea-poly(siloxane)-polyether product has better mechanical strength. The results show that the mild microphase separation between the soft segment aggregation phase and the hard segment aggregation phase can improve the mechanical properties of the polymer.

Aneja et al. [19] used polyurethane and polyurea as hard segments and polydimethylsiloxane as soft segments. The effects of soft segment length, hard segment type and content, molecular weight and symmetry of chain extenders on the morphology of the corresponded copolymers were investigated. Studies have shown that less content of polydimethylsiloxane soft segment in the copolymer produces more obvious microphase separation. In polyurethane and polyurea block copolymers with end-functionalized polydimethylsiloxane as soft segment, a greater degree of hydrogen bonding corresponds to a harder and firmer polyurea segment and a broader operating temperature range than the pure polyurethane with equivalent hard segment content.

In order to improve the tensile properties of polyurea copolymers only using polydimethylsiloxane (PDMS) as a soft segment component, Sheth et al. [20] incorporated poly (propylene oxide) (PPO) into polyurea copolymers in a controlled manner between PDMS and urea segments, and studied the effect of their solid structure on the properties. Because PPO has the ability of intersegmental hydrogen bonding with urea segments, it is selected as a copolymer soft segment. Dynamic mechanical analysis (DMA) demonstrated that the copolymers with PDMS as the soft segment only had a very broad and almost temperature-insensitive rubbery platform, and the incorporation of PPO segments resulted in a narrower and temperature-sensitive rubbery platform. Compared with PDMS-based polyurea-poly(siloxane) with an average molecular weight of 7000 g·mol^−1^, the tensile strength and elongation at break of the soft segment containing PPO with an average molecular weight of 2000 g·mol^−1^ were significantly improved.

Sirrine et al. [21] introduced in detail the method of synthesizing PDMS-polyurea block polymer by using urea and a disiloxane diamine chain extender in the molten phase. This process does not use isocyanate, is solvent-free and catalyst-free. The PDMS-polyurea obtained by melt polymerization can maintain optical transparency and good mechanical ductility. Differential scanning calorimetry (DSC) and DMA showed that the polymer had microphase separation. The results of tensile and hysteresis measurements confirmed that the properties of these PDMS-poly(urea) without isocyanate were similar to those of PDMS-poly(urea) with isocyanate. Therefore, this method of preparing high-performance elastomers without isocyanate has good commercial prospects.

In addition to silicone-modified elastomers with polyurethane and polyurea as hard segments, polyimide can also be added to poly(siloxane) block elastomers as hard segments to improve their physical and chemical properties. The modified poly(ethers) used in silicone block elastomers include poly(alkyl) ethers as soft segments and poly(aryl) ethers as hard segments. The block copolymer formed by poly(siloxane) and polyether is a mixed soft segment, and forms a block copolymer with other hard segment chains. Poly(aryl) ether contains a benzene ring, is not easy to rotate due to steric hindrance and serves as a hard segment to react with poly(siloxane) to generate silicone-poly(aryl) ether elastomer.

Andre [15] first synthesized a polyimide hard segment with aromatic dianhydride and aromatic diamine as the raw materials and allylamine as the capping agent, then synthesized polyimide-poly(siloxane) (PI-PHSX) block copolymer with poly(octyl dimethyl siloxane) (PHSX) as a soft segment and toluene as a solvent under the action of a Custer catalyst. Compared with oligomeric siloxanes based on -Si (CH_3_)_2_-O- units, PHSX has better thermal stability and better chemical resistance when used as a soft segment. Different PI-PHSX block copolymers with thermoplastic property can be obtained by only changing the hybrid siloxane segment, while keeping the allyl polyimide segment unchanged. The soft segment length has an effect on the thermal resistance, thermal degradation activation energy, mechanical properties and surface properties of PI-PHSX block copolymers with different PI contents. According to the activation energy calculation results of thermal degradation, the thermal stability of PI-PHSX block copolymer is improved when PHSX is used instead of traditional poly(siloxane) soft segment. In addition, PI-PHSX copolymers exhibit good thermomechanical properties and low surface tension.

#### 2.2.2. Epoxidation-Modified Elastomer

Most general rubbers such as natural rubber, styrene-butadiene rubber and isoprene rubber are diene rubber. In order to improve their performance and further expand their application, the double bonds in the diene rubber macromolecules can be chemically modified. Epoxidation modification is a simple and effective method [22]. Styrene thermoplastic elastomers, which contain diene units, can also be modified by epoxidation. In addition, there have been reports on the epoxidation of vinyl double bonds as the production capacity of silicone rubber containing double bonds has increased in recent years.

Epoxidation modification is the reaction of strong oxides, peroxides and unsaturated carbon-carbon double bonds in rubber, thereby introducing epoxy groups into the rubber molecular chain. The diene rubber or elastomer modified by epoxidation not only retains the structure and properties of the original material, but also enhances the intermolecular force due to the introduction of polar groups in the molecular chain, thereby increasing many new excellent properties, including oil resistance, good adhesion to other polymers and good compatibility with other materials. In addition, the epoxy group introduced in the rubber macromolecular chain can react with polyamines, carboxylic acids, anhydrides and other compounds to form a body structure, which provides a new green environmental protection method for the crosslinking of elastomers.

##### Performance and Application of Epoxidation Elastomer

Epoxidation changes the structure of diene rubbers or elastomers containing double bonds. Due to the introduction of polar and active epoxy groups, the properties of rubbers have greatly changed, and the reaction of epoxy groups can be used to develop more extensive applications. The main aspects are as follows.

Epoxidized elastomer has good wet skid resistance, oil resistance and low rolling resistance, which can be used in tires [23]. Cong et al. studied the application of epoxidized trans-1,4-polyisoprene (ETPI) in radial tire tread rubber, and found that the tread rubber had good wet skid resistance and low rolling resistance. Jacobi et al. [24] used epoxidized styrene-butadiene rubber (ESBR) instead of SBR to prepare thermoplastic vulcanizate (TPV), which significantly improved the oil resistance and solved the problem of poor oil resistance in traditional TPV.

Jiamjitsiripong et al. [25] found that ENR can promote faster curing, improve mechanical properties, the compression set and wear resistance of composites and significantly improve their gas barrier properties.

There are not only polar epoxy groups/segments, but also a certain amount of non-polar olefin segments in the molecular chain of epoxidized elastomers, thereby epoxidized elastomers still have good compatibility with other rubbers. Noriman et al. [26] studied the effect of ENR on the properties of SBR/NBR blends, and found that the processing properties of the blends became better after adding ENR, and properties such as modulus and tensile strength were improved. Wang et al. [27] added epoxidized Eucommia rubber (EEUG) to a SBR/silica composite system, which significantly improved the dispersion of silica in the rubber matrix, and improved the wear resistance, mechanical properties and compressive fatigue properties of the vulcanizates. Jiamjitsiripong et al. [25] added ENR as a compatibilizer to ENR/BIIR composites. The results showed that the addition of ENR made the filler more evenly dispersed, and the air tightness, wear resistance and mechanical properties of the composites were also significantly improved.

Epoxidized elastomer can be used as an interface modifier. Narathichat et al. [28] studied the modification effect of NR and ENR-50 on nylon-12, and found that the mechanical properties, thermal stability and stress relaxation behavior of nylon-12 were improved after adding ENR-50. This is due to the better compatibility of ENR-50 with nylon-12, resulting in good interfacial interaction and finer dispersion, forming a continuous phase structure in the blend.

Epoxidized elastomer can be used as a toughening agent. Wang et al. [29] prepared a bio-based thermoplastic vulcanizate (TPV) composed of polylactic acid (PLA) and ENR by dynamic vulcanization in the presence of dicumyl peroxide (DCP). Tanrattanakul et al. [30] compared the toughening effect of ENR and NR on nylon-6, and found that the tensile strength and yield stress of ENR- and NR-toughened nylon-6 composites decreased, and the elongation at break slightly increased. However, the impact strength of the composites increased after adding ENR, while the impact strength of the composites decreased after adding NR.

### 2.3. Nanocomposite Elastomer Materials

#### 2.3.1. Viscoelastic Polymer Materials

Polymer melts are both viscous and elastic. In polymer processing, viscosity response has been widely used as an index to evaluate the processing performance of polymers. However, only the viscosity of the polymer does not fully reflect the processing characteristics of the polymer. In fact, the viscoelasticity of the polymer really determines the processing characteristics of the polymer.

##### Study on the Theory of Extrusion Swell

Extrusion swelling is an important topic in non-Newtonian mechanics and polymer rheology. It has been shown that the melt cross-section size is larger than the die size when the polymer melt is extruded from the die. As early as 1893, Barus began to observe and study the phenomenon of extrusion swelling. In the initial period, macroscopic momentum conservation and energy conservation laws were mainly used to carry on the research. Metzner et al. [31] proposed a relationship between extrusion swelling and normal stress differences by using the law of conservation of macroscopic momentum.
(1)$${\left(P_{11}-P_{22}\right)}_{R,L}=\frac{\rho D_{0}^{2}}{64n}{\left(\frac{8V}{D_{0}}\right)}^{2}\times \left\{\left(n+1\right)\times \frac{3n+1}{2n+1}-{\left(\frac{D_{0}}{D_{e}}\right)}^{2}\left[n+1+\frac{dlogD_{0}/D_{e}}{dlog8V/D_{0}}\right]\right\}$$

P_11_ − P_22_ are the normal stress differences at the die exit, D_0_ is the die diameter, D_e_ is the diameter of the extrudate at the downstream, V is the downstream average velocity, n is the power flow index, and ρ is the melt density.

This relation is effective at high Reynolds number and relatively small elastic deformation, and is only applicable to polymer solutions. Mori and Han [32] modified the theory proposed by Metzner et al. and derived new correlations for polymer melts.

As early as 1948, Spencer and Dillon [33] believed that the extrusion swell of the die head was the elastic recovery of extrusion melt under stress in the flow channel. A. S. Lodge [34] proved in 1964 that when a micro-element of a rubber-like fluid suddenly transits from a shear stress state to a free relaxation state, there is an instantaneous elastic recovery, manifested as an expansion in the shear direction, followed by a smaller and slower recovery. Therefore, elastic recovery or elastic relaxation has become an important theoretical basis for explaining extrusion swelling.

Based on Lodge’s theory, Bagley et al. [35] proposed different expressions of extrusion swelling. Tanner [36] considered an infinite pipe flow. At a certain moment, the pipe wall was suddenly removed, and the fluid was instantaneously adjusted to a uniform flow with zero stress.
(2)$$B=D/D_{0}=0.1+{\left(1+S_{R}^{2}\right)}^{1/6}$$

Constant 0.1 is an empirical data; S_R_ is a recoverable shear strain.
(3)$$S_{R}=\frac{\tau_{11}-\tau_{22}}{2\tau_{12}}=\frac{\varphi_{1}\gamma }{2\eta }$$

γ is the shear rate. This result has nothing to do with the size of the flow channel, it only reflects the relationship between the extrusion swelling and the melt elasticity, and is applicable to the measurement of elasticity. In addition, only the history of shear flow is considered, which greatly simplifies the analysis of the process.

In the past decade, there has been a new research trend of introducing functional groups into the end or side groups of linear rubber macromolecules, and assembling macromolecules into supramolecular elastomers with a network structure through non-covalent bonds between functional groups, such as hydrogen bonding, coordination bonds, ionic bonds, etc. Such supramolecular elastomers exhibit properties similar to those of thermoplastic elastomers: a loss of crosslinking due to the dissociation of non-covalent bonds at high temperatures, plastic flow of rubber macromolecules, easy processing and reusability and multiple processing. Among them, research on the preparation of supramolecular elastomers using hydrogen bonding has attracted the most attention. In recent years, there has also been a new direction of introducing multiple hydrogen bonds between small molecules to prepare hydrogen-bonded supramolecular elastomers through self-assembly between hydrogen bonds, and has gradually developed into a new research field.

#### 2.3.2. Supramolecular Elastomer Based on Hydrogen Bond Self-Assembly between Macromolecules

According to the concept of supramolecular proposed by Lehn, the 1987 Nobel laureate in chemistry [37], supramolecules are defined as organized entities with higher complexity formed by the association of two or more chemical species through intermolecular non-covalent bond forces. On this basis, Stadler and his collaborators [38] proposed the use of direct hydrogen bonding between macromolecular chains to prepare thermoplastic elastomers: based on polybutadiene, a series of polybutadiene thermoplastic supramolecules were synthesized by changing the hydrogen bond acceptor/donor pair at the end of the macromolecular chain (acceptor: A, donor: D, the two form an A/D pair, referred to as hydrogen bond synthon); for example, 4-phenyl-1,2,4-triazoline-3,5-dione (PTD). The presence of hydrogen bonds was confirmed by FT-IR. As the temperature increased from 40 °C to 80 °C, the characteristic absorption peak of the carbonyl group gradually shifted from 1701 cm^−1^ to 1723 cm^−1^, indicating that the hydrogen bond strength was gradually weakened.

In 1997, Meijer et al. [39] developed an important class of AADD assembly units with quadruple hydrogen bond interactions: 2-ureido-4[1H]-pyrimidinone derivatives (UPy), and grafted them with UPy units at the end of ethylene-butene copolymer (PEB) macromolecular chains to prepare PEB-UPy supramolecular polymers [40] (as shown in Figure 3). Compared with ungrafted PEB, PEB-UPy has higher mechanical strength and exhibits typical thermoplastic properties at room temperature.

Compared with grafting hydrogen bond synthons at the end of macromolecular chains [38], grafting hydrogen bond synthons on the side groups of macromolecular chains can increase the average concentration of hydrogen bond synthons on each macromolecular chain, and thus, it is more important [40]. Chino grafted 3-amino-1,2,4-triazole (ATA) onto the macromolecular chains of polyisoprene (IR), butyl rubber (IIR) and ethylene propylene rubber (EPM) via maleic anhydride to prepare hydrogen bond crosslinked IR-*g*-ATA, IIR-*g*-ATA or EPM-*g*-ATA with different grafting rates. It was found that the tensile strength significantly increased with the increase in ATA grafting rate, as shown in Figure 4 [42]. Similarly, Chang et al. grafted ATA onto the side groups of EPM and polyethylene (PE) by maleic anhydride, and prepared EPM-ATA/PE-*g*-ATA blends that could be repeatedly processed; the processing properties remained basically unchanged.

Peng [43] prepared polybutadiene (PB) side-linked sulfonyl isocyanate (SU) (PB-SU) by a three-step reaction, as shown in Figure 5. When the grafting rate of SU (relative to the double bond content of PB main chain) is more than 1%, PB-SU exhibits hydrogen bond crosslinking characteristics, but the maximum grafting rate is only 4%. Cheng et al. [44] introduced 3-amino-5-acetamide-1,2,4-triazole (AATA) and 4-aminouracil (AU) to partially replace the bromine atoms of brominated isobutylene-p-methyl styrene copolymer elastomer (BIMS) and brominated butyl rubber (BIIR) in the presence of a phase transfer catalyst, respectively. AATA and AU were introduced into the polymer chain, and polyisobutylene networks (BIMS-AATA and BIMS-AU) and thermo-reversible crosslinked butyl rubber (IIR-AU) were formed by the self-assembly of intermolecular hydrogen bonds. However, the maximum mass fraction of hydrogen bond groups in the two products is low due to the limitation of macromolecular chain curl and low two-phase reaction efficiency.

#### 2.3.3. Graphene/Elastomer Nanocomposites

Uncrosslinked rubber, like liquid, cannot withstand external loads. Crosslinking results in the formation of a permanent polymer network, which gives the rubber a solid-like elastic property. Traditional elastomer processing requires the use of specific chemical reagents to crosslink the elastomer and use filler particles to enhance it.

Elastomer nanocomposites are mainly composed of three main parts: a rubber matrix, a filler network constructed by nanoparticles as a reinforcing agent, a chemical network composed of crosslinking agents such as sulfur or peroxide, and auxiliary systems such as an antioxidant, a plasticizer and a softener. In these components, the filler network composed of nanoparticles has the greatest impact on the overall performance of rubber composites. The main factors affecting the filler network include the physical and chemical properties of the filler particles themselves, the filler content, their dispersion in the rubber matrix, and their interaction with the rubber molecular chain. Traditional nanofillers such as carbon black, silica white still play a dominant role in the global rubber industry. However, with the improvement of rubber products performance requirements, a series of new fillers such as short fibers, carbon nanotubes and nano clays have also emerged in recent decades. These fillers have certain characteristics in terms of structure (such as tubular, rod, flake, etc.), and have a significant effect on some rubber properties (such as 100% elongation, tear resistance, electrical conductivity, etc.). Further combinations with carbon black or silica have been adopted to enhance rubber and improve the performance of rubber products.

Nanoparticles can enhance the properties of elastomers through their own surface properties after polymerization modification with elastomers. Donnet [45] studied the effect of the filler surface and proved the existence of active sites in the case of carbon black. The surface energy is proportional to the specific surface area of the filler, which also indicates the importance of the interaction between the surface and the elastomer chain. The surface defects of the dispersed phase with small particle size are relatively few, and there are many unpaired atoms. The ratio of the number of surface atoms to the total number of atoms of nanoparticles sharply increases with the decrease in particles. The crystal field environment and binding energy of surface atoms are different from those of internal atoms, which have great chemical activity. The particle of the crystal field and the increase in the active surface atoms greatly increase the surface energy, so it can be closely combined with the polymer substrate, and the compatibility is better. When subjected to external force, the ion is not easy to separate from the substrate, and can better transfer the external stress. At the same time, under the interaction of the stress field, more micro cracks and plastic deformation will be generated inside the material, which can cause the substrate to yield and consume a lot of impact energy, so as to achieve the purpose of strengthening and toughening at the same time. Nano-alumina is widely used in thermal conductive plastics, thermal conductive rubber, thermal conductive building age, thermal conductive coatings and other fields because of its good insulation and thermal conductivity. Graphene (GE) is a two-dimensional sheet structure. The basic structural unit is a sp^2^ hybrid hexagon composed of C atoms, and the thickness is a single atomic layer or multiple atomic layers. The theoretical specific surface area of GE is 2630 m^2^/g, the Young’s modulus is 1100 GPa, the thermal conductivity is 5300 W/(m·k) and the electron mobility is 15,000 cm^2^/(v·s) [46]. Due to its unique structural characteristics, it has quickly become a hot spot in many research fields such as energy, electronics and materials. In the field of elastomers, many researchers have used GE to prepare high-performance GE/elastomer nanocomposites. GE has unique advantages in elastomer enhancement, dynamic and static performance, as well as electrical and thermal conductivity. Xing et al. [47] reported a method for simultaneously crosslinking and reinforcing styrene-butadiene rubber (SBR) with graphene oxide (GO). We found that graphene oxide is not only an effective reinforcing filler, but can also produce free radicals when heated, so that SBR has covalent crosslinking. In addition, the interaction between the graphene oxide surface and the SBR polymer forms an interface layer, in which the crosslinking density increases to the graphene oxide surface, so the interface layer exhibits a much slower relaxation dynamic than the bulk rubber. The unique role of graphene oxide gives graphene oxide/SBR nanocomposites better mechanical properties than SBR crosslinked with traditional sulfur or dimethyl peroxide.

### 2.4. Self-Healing Elastomer Materials

#### 2.4.1. Supramolecular Elastomer Based on Hydrogen Bond Self-Assembly between Small Molecules

The preparation of thermo-reversible supramolecular polymers with a network structure by hydrogen bond interaction between small molecules (or oligomers) has been a hot topic in the field of supramolecular chemistry in recent years [41]. Theoretically, when the two ends of the small molecule have hydrogen bond synthons that can interact with each other under certain conditions, “long-chain polymers” can be formed due to the interconnection of hydrogen bonds. When the average number of hydrogen bond functional groups of the small molecule is greater than two, it may also form a supramolecular structure with a network structure connected by hydrogen bonds. The process of forming “macromolecules” (supramolecule) through hydrogen bond interaction between molecules is apparently very similar to the condensation polymerization of small molecule monomers, as shown in Figure 6.

After Meijer et al. proposed the UPy connection unit, the study of preparing supramolecular polymers by connecting UPy and its derivative groups or similar multiple hydrogen bonding groups at the end of small molecules has rapidly developed and a variety of other types of quadruple hydrogen bonding connection units have been synthesized [40]. For example, PDMS-UPy supramolecular elastomers with viscoelastic properties similar to PDMS at room temperature were prepared by connecting UPy units at the end of dimethyl siloxane oligomers. It should be noted that the UPy unit is grafted at the end of the linear small molecule (oligomeric siloxane). Although the small molecule can be assembled into a “long chain” supramolecule through multiple hydrogen bonds, these “long chain” supramolecules are not effectively crosslinked due to the long chain molecules. It is difficult to show the basic properties of the elastomer in the general sense: it has good high elasticity at room temperature and can quickly return to its approximate initial shape or size after the external force is removed [1].

The study of St Pourcain and Griffin [48] also found that the terminal group or side group of small molecules (or oligomers) is often a hydrogen bond donor/acceptor functional group with the same structure, making small molecules easy to form a more regular arrangement; that is, to form crystals, when the hydrogen bond between small molecules is strong. The prepared supramolecular network often exhibits a partially crystalline plastic shape at room temperature, or even a fibrous state, rather than an elastomer. Leibler et al. grafted a variety of amide-based synthons with multiple hydrogen bonds at the end of dimer acid small molecules (including monocarboxylic acid, dicarboxylic acid and poly(carboxylic) acid) as shown in Figure 7, which are translucent glassy semi-brittle materials at room temperature. When heated to 60–90 °C, they can exhibit typical rubbery elasticity, and their deformation can be almost completely restored, with ultra-low hysteresis characteristics that traditional covalently crosslinked rubber does not have; another surprising feature of this supramolecular elastomer is that it can self-heal after cutting at room temperature [49], as shown in Figure 8, and its strength after healing increases with the prolongation of healing time. At high temperatures, hydrogen bonds in supramolecular elastomers dissociate, and the elastomer becomes a small molecule liquid with extremely low viscosity, which makes the molding process of supramolecular elastomers extremely convenient. After this work was published in Nature, it received extensive attention in the field of supramolecular research [49], known as the “new generation of elastomer materials”. It provides a good start for the study of supramolecular elastomers based on hydrogen bonding between small molecules, opening up a new research field of supramolecular elastomers based on the multiple hydrogen bonding of small molecules.

#### 2.4.2. Self-Healing Polyurethane Elastomer

The introduction of self-healing functional groups in the structure of polyurethane elastomer, using the reversible reaction of its internal molecular structure or macromolecular diffusion, is expected to form self-healing elastomer. Self-healing polymers based on dynamic reversible systems include disulfide bonds, Diels-Alder, imine bonds, hydrogen bonds and metal coordination [50]. Among them, based on the reversible covalent interaction of disulfide substitution reaction, disulfide bonds can activate the dynamic association or dissociation of dynamic bonds through moderate stimulation (including heat, light and pH) [51], thereby achieving self-healing of the material. However, the dynamic properties of disulfide bonds not only endow the material with self-healing properties, but also reduce the strength of the material, which is also a problem that this type of self-healing material has been urgently needed to solve.

Liu et al. [52] used aliphatic disulfides to prepare self-healing and recyclable crosslinked poly (thiocarbamate-carbamate). The tensile strength of the thermosetting elastomer obtained from asymmetric diisocyanate was 1 MPa. After 48 h in natural light, the self-healing efficiency reached 85.6%. Kim et al. [53] reported an aromatic disulfide polyurethane elastomer with easy processing and rapid self-healing at room temperature. Its tensile strength was 6.76 MPa, the elongation was more than 900% and the self-healing efficiency reached 88% after 2 h at room temperature.

Gao et al. [54] further prepared polyurethane elastomers based on polysulfides. After the tensile strength reached 5.8 MPa at 75 °C for 24 h and 100 °C for 4 h, the self-healing efficiency reached 90.8% and 93.1%, respectively. Moreover, it was found that the flexibility of polysulfide segment molecules and low crosslinking degree were beneficial to enhance the repairability of the sealant network. It can be seen that the repair efficiency and strength of disulfide self-healing materials cannot be simultaneously improved at present.

Jian et al. [55] prepared a self-healing adhesive based on aside glycidyl ether in the early stage, with a tensile strength of 1.03 MPa and a self-healing efficiency of 98.2% at 60 °C for 24 h. In order to accelerate the self-healing rate, a disulfide-type poly(tetrahydrofuran) polyurethane was further prepared [56]. When the mass fraction of disulfide reached 6%, the tensile strength of the elastomer was 5.01 MPa, and the self-healing efficiency at 60 °C for 6 h reached 100%, which was higher than that of the previous material.

In order to further improve the strength of this type of material and coordinate the contradiction between the strength and self-healing properties of the elastomer, a polyurethane elastomer with high strength and high self-healing efficiency was prepared by prepolymer method using a polymer-type disulfide crosslinking agent. Based on the investigation of self-healing efficiency under different structures and different conditions, the hardness and thermal stability of polyurethane elastomers were studied.

##### Elastomer Self-Repair Process

The elastomer was cut off and placed in an oven at 80 °C for 24 h. When it was taken out and a tensile test was performed, it was found that the sample was not broken even if the necking phenomenon occurred. This process is shown in Figure 9.

##### Effect of Elastomer Structure on Self-Healing

In order to investigate the effect of the elastomer structure on self-healing properties, the elastomer with cracks was repaired at 80 °C for 24 h, and the tensile strength was compared with that of the original sample without cracks to calculate the self-healing efficiency. The results are shown in Figure 9 and Table 2.

As shown in Figure 10, with the increase in the amount of disulfide crosslinking agent, the tensile strength of the system increases first and then decreases; PUSS-5 had the highest tensile strength of 11.90 MPa. The tensile strength of PUSS-6 was only 7.88 MPa. This was related to the larger molecular structure of the crosslinking agent used in the system. As the amount of crosslinking agent increases, the crosslinking density of the system increases; consequently, the tensile strength increases. However, with the further increase in crosslinking agent, the regular arrangement of soft segments is limited, and the mass fraction of disulfide also increases, resulting in a decrease in tensile strength due to the bond energy of a disulfide bond(S-S) (about 240 kJ/mol) being lower than that of a carbon-carbon bond (about 350 kJ/mol). The elongation at break of the elastomer also has the same variation trend. When the mass fraction of disulfide bond is low, the molecular chains in the system are relatively independent and it is easy to produce relative slip, and the elongation at break is high because a certain number of free chains do not participate in the crosslinking reaction. The elongation at break of PUSS-4 reaches 559%. When the mass fraction of disulfide increases, the three-dimensional network structure restricts the molecular chains to each other and prevents relative slip, thus reducing the elongation at break.

It can be seen from Table 2 that, with the increase in the mass fraction of disulfide, the self-healing efficiency of the elastomer significantly increased from 54.5% of PUSS-3 to 99.5% of PUSS-6. It is believed that the occurrence of self-healing was due to the appearance of many open dynamic disulfide bonds on the fracture surface. Under external stimuli, the bonding is re-established and the mechanical integrity of the fracture interface is restored. At the same time, it was also found that the excessive amount of disulfide crosslinking agent would lead to a decrease in the mechanical properties of the elastomer. Compared with the tensile strength of PUSS-5 of 11.9 MPa, the tensile strength of PUSS-6 was decreased to 7.88 MPa, indicating that the disulfide structure was beneficial to the self-healing of the system, but it was not conducive to the improvement of the strength of the elastomer. Although the disulfide bond self-healing system is a complexation covalent adaptive network, the network topology rearrangement occurs during self-healing, and the network structure can still remain intact, but excessive disulfide content is not the optimal choice.

##### Effect of Temperature on Self-Healing

Self-healing efficiency is not only related to the structure of the elastomer itself, but it is also directly related to the external conditions (temperature and time). Therefore, the self-healing efficiency of PUSS-6 elastomer after 24 h at different temperatures (25, 40, 60, 80 °C) was investigated. The results are shown in Figure 11 and Table 3. The tensile stress-strain curve in Figure 11 is divided into three stages. In the initial stage, the tensile strength significantly increases, which is caused by the stretching of random coil segments in the elastomer (elastic deformation). In the second stage, the stress slowly increases after the yield point. However, the deformation caused by the extension of the soft segment of the elastomer is very large; that is, strain softening occurs, and the curve shows a gentle increase. In the third stage, the tensile stress-strain curve begins to sharply increase again, which is related to the orientation of the soft segment in the elastomer and the deformation of the hard segment of the polyurethane under high stress.

From Table 3, it can be seen that the self-healing efficiency of the elastomer at 25 °C is 31.3%; when the temperature rises to 80 °C, the self-healing rate reaches 99.5%. This phenomenon occurs because the heating accelerates the movement rate of the chain segment, and the movement of the chain segment promotes the rapid replacement of the reversible bond, so as to realize the repair of the damaged crack, and even reintegrate through the broken interface, penetrate each other and realize the recovery of material strength.

Further analysis of Figure 10 is shown in Table 4. As can be seen from Table 4, the elastic modulus of the elastomer increases with increasing self-healing temperature. This is because heating makes the elastomer reaction degree further improved; at the same time, the temperature rise makes the polymer chain segment move faster and it is more easily arranged, and then the strain decreases under the same stress, showing an increase in elastic modulus. The yield stress of the elastomer obviously changes with the self-healing temperature. Generally, the yield stress decreases with the increase in heat treatment temperature. However, due to the cracks in the elastomer in this paper, the self-healing rate of the sample is low at a lower temperature. Therefore, the yield stress increases with the increase in self-healing temperature, showing a completely opposite change in trend.

##### Effect of Time on Self-Healing

At 80 °C, the tensile test of repaired PUSS-6 elastomer after different times was carried out, and the results were shown in Figure 12 and Table 5. It can be seen that the self-healing efficiency increases with the extension of self-healing time. After 2 h of self-healing, the hydrogen bond at the notch has not been fully formed in a short time, and the disulfide bond cannot be completely extended, resulting in an uncomplete dynamic replacement reaction. The tensile strength of the elastomer is 3.81 MPa, and the self-healing efficiency is only 48.4%. After 6 h, the repair groups take obvious effect. Among them, the bond energy of the hydrogen bond is lower than the disulfide covalent bond, and thus the rate of fracture and formation is greater than the disulfide bond. Therefore, the hydrogen bond plays a reversible interaction, which significantly promotes the self-healing efficiency. The tensile strength increases to 5.74 MPa, and the self-healing rate reaches 72.8%. With the further extension of time, the dynamic reversible exchange reaction of the disulfide bond continues to play a major role, and the growth rate of self-healing efficiency slows down. The dynamic reversible reaction of disulfide bonds and hydrogen bonds has been basically balanced, and ultimately achieves 99.5% self-healing efficiency until 24 h. Combining with the effect of temperature on self-healing efficiency, self-healing also conforms to the principle of time-temperature equivalence. Increasing temperature and prolonging time are equivalent to the movement of polymer chains.

### 2.5. Rubber Elastomer Material

#### 2.5.1. Fluorine Elastomer Material

Fluorine elastomer is the rubber that hydrogen atoms in the carbon main chain polymer are substituted by fluorine atoms, which has excellent thermal resistance, oxidation resistance, solvent resistance, and has good tensile properties and compression permanent deformation properties, and belongs to a high-performance elastomer.

##### Fluorine Elastomer Classification [57]

(1) Fluor rubber (FKM)

FKM is the most diverse and abundant type of fluorine elastomer (over 80%). All FKM fluorine elastomers contain vinylidene fluoride (VF_2_). In the D1418 standard, FKM is defined as polyvinylidene fluor elastomer, using vinylidene fluoride as a comonomer, with a substituted fluorine, alkane, perfluoro hydrocarbon or perfluoro alkoxy group in the polymer chain, with or without a vulcanization point monomer (having a reaction side group). To date, there are five kinds of FKM elastomer, and the specific monomer composition is shown in Table 6. FFKM is polymethyl perfluoro rubber, and all substituents of the polymer chain are fluorine, perfluoroalkyl or perfluoro alkoxy groups. FEPM is a poly(methylene) fluor elastomer containing one or more monomeric alkyl, perfluoroalkyl and perfluoro alkoxy groups, with or without the curing point monomer.

Properties of Fluorine Elastomer: The research shows that with the increase in fluorine content, the methanol resistance of fluorine rubber is obviously improved and the fuel transmittance is greatly reduced, but the low temperature resistance is also reduced, as shown in Figure 13 and Figure 14; TR10 is the temperature corresponding to the shrinkage percentage of 10%.

Main Application Fields of Fluorine Elastomer:Fluorine elastomer has been widely used in liquid sealing and isolation applications in aviation, automation, chemistry, petrochemistry, food and drug processing, construction and other industries, manufactured into various components such as diaphragms, O-rings, gaskets, building coatings, pipelines, valve seals and shaft seals.
The General Application Fields of Fluorine Elastomer

The partial aplications of fluoroelastomers are shown in Table 7 and distribution of fluoroelastomer items are shown in Figure 15.
2.Application of Fluorine Elastomer in the Petroleum and Natural Gas Industry

Fluorine elastomer has been applied in the oilfield industry. However, the requirements for fluorine elastomers will continuously increase as wellbore conditions, such as temperature, pressure and chemical media, become more severe. APA (Advanced Polymer Alloys) fluorine elastomer can be used to prepare rubber products with better processing performance, better quality and a more complex shape.

A-type FKM is the most commonly used polymer in the oilfield industry. It has a good balance in chemical resistance, high temperature and room temperature physical properties. The methanol resistance of GF-S with high fluorine content was improved in comparison. FEPM (ETP-S and TFE/P) have better amine preservative-resistant performance, but their physical properties at high temperatures are poor, and TFE/P has poor elastic properties at low temperature. The selection of fluorine elastomer in different environments can refer to Table 8.

### 2.6. Expansive Elastomer Materials

Expandable elastomer is a new type of advanced polymer that expands when interacting with liquids such as water, oil or acid. Expansion causes changes in geometry, density, hardness and other properties [58,59,60]. Water-expanded elastomers absorb saline solution and expand through a permeation mechanism, while oil-expanded elastomers absorb hydrocarbon and expand through a diffusion process [61]. The swelling ratio depends on the temperature, pressure, elastomer type and fluid composition. Expandable elastomers have become common materials for some petroleum applications, such as interlayer isolation. A typical expandable packer structure is shown in Figure 16. By employing different expansion elastomers, the profitability of old wells can be maintained, abandoned wells can restart production and new reservoirs that are difficult to exploit can be economically produced.

The automatic operation mechanism of the expansion packer makes installation and execution easier, minimizing significant drilling time and associated costs. Some case studies on the deployment and implementation of expandable elastomers in more important oil and gas applications will be discussed in the following sections.

(1) Regional isolation

Interzone isolation refers to the technology used to prevent the mixing of excess fluid and production fluid in improper areas. In general, interzone sealing in the wellbore is achieved by cementing the production string and appropriately using casing plugs and packers. Figure 17 shows the free expansion of the elastomer during effective interzone isolation. The Asab oilfield in Abu Dhabi is a mature carbonate reservoir. It was drilled in 1985 and sidetracked in 1999. By 2005, water content increased from 14% to 25%, significantly reducing oil production [62]. Due to the placement failure of the cement plug at the toe, water flows out of the crack. The expandable packer is used to isolate unwanted layers, and the water content is greatly reduced from 25% to 0.3%. In the South Furious oil field in Malaysia, the deployment of the expandable elastomer reduced the water content and began production the next day, even before the elastomer completely expanded [63]. Enhanced oil recovery (EOR) technology is used to obtain higher recovery from old reservoirs that the production ability of which gradually reduced or new reservoirs difficult to access. The required oil, gas and water isolation can be successfully achieved through expansion elastomers and one-time cementing. The Oman Petroleum Development Department implemented the first EOR plan in the Harweel Cluster block in southern Oman, using miscible gas injection. In order to improve oil and gas recovery, it is necessary to properly isolate the production zones. In many applications, cementing failures have led to the use of expandable elastomers and very good results have been achieved. Other important applications of expandable elastomers for interzone isolation include production water management, sand control, reservoir division, production separation, inflow profile control and condensate oil migration control.

(2) Well completion

In petroleum applications, all operations performed before the well is ready for production are referred to as well completion. Drilling, installing/cementing the casing, continually drilling until the desired depth is reached, then well completion is accomplished by perforating/boosting the casing, cement and formation. The purpose of cementing is to provide effective hydraulic seals between the casing and the formation to prevent annular flow and isolate individual intervals. By installing unique automatic or manual intervention completion and monitoring tools, expandable elastomers can provide effective interlayer isolation that conventional perforation and cementing techniques cannot achieve, thereby reducing development costs and optimizing production, which is one example of an intelligent well. In Qatar’s Al-Khalij offshore oil field, corroded and damaged strings were repaired using the smart well method [64]. Before starting the re-completion, the alternative packer technology was studied. Reliable hydraulic sealing mechanisms can provide technically and economically feasible workover for such mature reservoirs. The expandable elastomer packer is used to prolong the life of the well and improve the recovery efficiency. Expandable elastomers can also be used as completion tools for open holes, cased holes, horizontal wells and solid expandable tubular (SET) technology [65].

(3) Production stimulation

In the production layer of the well, it is sometimes necessary to expand the original channel or open up new channels through acidification, formation fracturing and other technologies. Such processes are called stimulation. Fracturing is a method of production stimulation. By opening new flow channels in the rock structure around the production well, can it increase the surface area of the formation fluid flowing into the well, and extend to any possible fractures near the wellbore. The application of expansion elastomer packers in production operations is shown in Figure 18. The main challenges are downhole environmental conditions, such as temperature and pressure, pipeline shrinkage and thermal effects originating from sealing shrinkage (which may cause temperature drops due to contact with the stimulation fluid). An American operation company used an expandable liner hanger, an expandable elastomer packer and a ball sliding sleeve to increase production in the open hole completion of horizontal wells, and achieved a total of 169 production increases. Expandable elastomer technology has also achieved success in the fields of hydraulic fracturing, matrix acidification and multi-stage fracturing.

(4) Underbalanced drilling

During the underbalanced drilling (UBD) of oil and gas wells, wellbore pressure is deliberately maintained below the formation fluid pressure, as shown in Figure 19. It allows the formation fluid to rise to the ground during drilling to prevent formation damage. UBD technology is used to determine the possible leakage area so that the expansion packer can be correctly placed [66]. The combination of UBD and expandable elastomer can provide an effective interlayer isolation mechanism. It can help maximize oil well performance and ultimately improve oil and gas recovery. Richard et al. explained the reason for using underbalanced drilling technology in the Nimr reservoir in Oman [67]. UBD inversion data show that free fractures and rip currents play a crucial role in water movement. Expandable elastomers are used for water plugging identified by UBD.

(5) Evaluation of an expansion packer

An expandable packer is a new technology to replace conventional technology in various petroleum applications. Accordingly, there should be some ways to evaluate the performance of these new applications. Cable ultrasonic measurement is used to evaluate the interlayer isolation achieved by expandable packers. Herold et al. discussed the evaluation of the interlayer isolation of expansion elastomer tools using conventional and latest ultrasonic tools. Ultrasonic imaging tools and ultrasonic scanners were used to measure acoustic impedance (AI), third interface echo (TIE), radius and bending attenuation. In the case of expandable elastomers, TIE offers more promising results in guaranteeing hydraulic sealing and interlayer isolation. Titanium content is generally measured from the cement stratum boundary. TIE can also be divided into swelling and inert elastomers [68]. Expansive elastomer is a developing technology with great potential. However, many challenges remain to be overcome. Applications in high pressure and high temperature (HPHT) extreme well environments are still questionable. Reservoirs with higher acidity and stronger chemical reactivity are also unexplored areas. The development of novel expansion elastomer materials, design improvements and innovative applications are urgently needed for enveloping high temperatures and high pressures, as well as erosive reservoirs.

#### 2.6.1. Water-Absorbing Expandable Elastomer Material

Water swellable polyurethane elastomer (WSPUE) is a new type of sealing polymer material with both elastic sealing and water swelling. It maintains the high elasticity like rubber and has the performance of rapid water absorption. It is mainly used as a civil construction waterproof material, filler, sealing material, ground building and underground pipeline waterproof material [69].

Water swellable polyurethane elastomer is prepared by using hydrophilic polyether polyol as soft segments and a diisocyanato chain extender as hard segments. The reason why the material can swell in water is mainly because the polyether molecular chain has hydrophilic units that absorb water. After absorbing water, the polyether chain segments change from a highly entangled state to a chain stretching state. The soft segment and the hard segment are connected by chemical bonds, and the molecular structure is relatively stable. After soaking in water, it will not precipitate from the matrix. After long-term or repeated soaking, its expansion rate and mass loss remain basically constant. After drying, the overall structure of the material does not change much and can be reused well. Therefore, the material has been rapidly developed as a waterproof material, and has broad application prospects.

##### Underlying Mechanism

Usually, the elastomer matrix is mainly composed of a hydrocarbon chain of high degree polymerization, and is hydrophobic. However, after the introduction of hydrophilic groups or components in the matrix, water molecules will enter the matrix once in contact with water molecules, and form a strong affinity with the hydrophilic groups in the elastomer. The hydrophilic material in the matrix is dissolved or swelled. Osmotic pressure difference is formed, promoting the penetration of water into the interior of the elastomer network. The hydrophilic substances substantially absorb water and cause the deformation of the matrix. When the deformation resistance of the elastomer and osmotic pressure difference achieve equilibrium, the maximum expansion rate of the hydrostatic is achieved, and the water swelling effect remains relatively stable. Studies have suggested that there are two forms of water absorption process; one is through capillary adsorption and diffusion; the other is through hydrogen bonds to form bound water.

WSPUE polyether polyol has a long chain segment of ethylene ether. The oxygen atom in the chain segment (-CH_2_-CH_2_-O-) has two pairs of lone electron pairs, which can be connected with the hydrogen atom in the water molecule to form hydrogen bond association. Due to the formation of the hydrogen bond, the molecular chain of polyethylene oxide changes from zigzag to tortuosity. The oxygen atom in the hydrophilic ether bond is located on the outer side of the chain, and the hydrophobic 1, 2-ethylene is located on the inner side of the chain. Therefore, the polymer becomes easy to combine with water and cause volume expansion of the material. The polymer with this molecular chain segment is zigzag in an anhydrous state, and becomes tortuous after absorbing water. The structure is shown in Figure 20 [70].

#### 2.6.2. Oil-Absorbing Expandable Elastomer Material

##### Expanded Graphite Material

Expanded graphite is a worm-like carbon material with a porous and high-specific surface (as shown in Figure 21) formed by the deep processing of flake graphite. It is recognized as one of the most valuable and promising products in carbon materials. Expanded graphite maintains the microscopic molecular structure of graphite materials, and has many excellent properties such as high and low temperature resistance, corrosion resistance and is self-lubricating. Due to its light weight, abundant pores and soft texture, expanded graphite has unique properties such as adsorption, compression resilience and sealing. Today, expanded graphite has been widely used in environmental protection, electrical power, chemical catalysis, machinery, the military and other fields [71].

The strong adsorption capacity of expandable graphite has been unanimously recognized all over the world. It shows good application prospects in environmental protection fields such as the emergency treatment of marine oil spill accidents. However, there are still some deficiencies in the field application of expandable graphite. The low hardness and tensile strength and brittle properties make it prone to deformation and crushing. At present, the energy consumption of the preparation process is high, and a large amount of strong acid wastewater will be produced during the preparation process. It is necessary to further explore more green and efficient preparation and regeneration methods. The adsorption mechanism of some substances is not clear, affecting large-scale application. The preparation, performance and application of research of the composite materials of expandable graphite and other materials is not enough. Most research is carried out in a laboratory, and lacks a pilot test and practical engineering application examples.

In view of some shortcomings of the abovementioned expandable graphite, in-depth research should be carried out from the following approaches. Developing new techniques to study the process of adsorbing specific substances by expanded graphite and the adsorption mechanism, and analyze the internal relationship between adsorption and desorption processes to achieve the process control of adsorbing specific substances. To improve the mechanical hardness of expanded graphite, prolonging service life, reduce the cost of expanded graphite preparation. Further exploring the migration and transformation process and migration mechanism of pollutants in the regeneration process, and seeking green and environmentally friendly regeneration methods. With in-depth study of expandable graphite and the figuring out of the above problems, expandable graphite will be widely used in trace oil adsorption materials, metal/non-metallic composite materials, sound-absorbing materials, medical materials and related fields.

## 3. Application of Elastomer in Drilling Fluids

### 3.1. Elastomer Plugging Material

Drilling fluid loss refers to the uncontrolled leakage of drilling fluid into the formation. According to its rate, fluid loss is divided into seepage loss (<1.6 m^3^ h^−1^), partial loss (1.6–80 m^3^ h^−2^) and complete loss (80 m^3^ h^−1^ or no return to the ground) [72]. Under the seepage loss, the drilling operation is normally carried out, but when high loss or complete loss occurs, the drilling operation must be interrupted [72]. The time spent on lost circulation control is uncertain and may be substantial, leading to significant drilling costs [73]. In addition, some severe events may occur after lost circulation, such as wellbore instability, pipe sticking and blowouts [74]. If it occurs in the oil producing area, fluid loss usually leads to reservoir damage and production decline [75]. Well leakage control is designed to minimize drilling fluid losses and ensure that drilling fluid returns to the ground [76]. Therefore, it plays a vital role in the research and application of drilling fluid plugging agent.

#### 3.1.1. Application of Elastomer Plugging Material

##### Application of Underground Crosslinked Elastomer Plugging Material

There are mainly cement slurry, bridge plugging and polymer gel plugging materials for fracture loss circulation and fractured/caved loss circulation. However, elastomeric materials have certain applications in plugging due to their unique properties.

Feng et al. [77] developed downhole crosslinked elastomer plugging material. This kind of plug agent was consistent with emulsion, phase inversion agent and crosslinking agent. It could form a high strength elastomer when the plug agent was pumped into the leaking layer and mixed with a certain amount of water. The elastomer with strong ability could offer good resident ability to reduce cement slurry thickening by underground water and plugging mud, and increased the ratio of success. The experimental results showed that crosslinking time was not affected by temperature, salt water, drilling cuttings, bentonite mud and other pollutants. The plug agent has been successfully used in several wells of the sixth block in Sudan.

Skov et al. [78] invented a downhole elastomer plugging combination agent, which could cut off the backflow and allow the controlled transport of substances to a predetermined location. In addition, during offshore oil recovery treatment, it could also ensure that the oil recovery equipment in the water injection well had good stability. At the same time, this elastomer plugging agent could also provide a composition that is easily pumped at lower temperatures and rapidly solidified at higher temperatures to provide elastomer plugging.

##### Application of Polyurethane Elastomer Plugging Material

Polyurethane elastomer material is also a good plugging material, and has been widely used in the field of plugging.

Wu et al. [79] studied a new type of mine polyurethane elastomer sealing plugging material, using polyurethane elastomer as a sealing plugging material, with a tensile strength of 2.45 MPa and an elongation at break of 610%. New mining polyurethane elastomer sealing plugging material has excellent physical properties, an adjustable range and can meet the needs of different environments; it has good tear strength and tensile strength, and can deform with the deformation of coal and rock mass without cracks. Strong coating adhesion, a variety of surfaces have excellent adhesion; curing time can be adjusted according to the construction needs after curing the surface to form a complete jointless elastic coating; the coating has good compactness and does not produce any leakage points.

Drilling fluid leakage into fractures is a challenging problem, which can lead to high-cost drilling and formation damage. Lashkari et al. [80] used conventional LCM to seal wide fractures, which increased the risk of bit nozzle clogging. Thermoplastic SMPU LCMs with a molar ratio of 21: 20: 1 of HDI, BDO and PCL were synthesized by two-step repolymerization using PCL with a molecular weight of 22711 (g·mol^−1^) as a soft segment. Infrared spectrum results showed that SMPU was completely synthesized. X-ray diffraction results showed that the crystallinity of SMPU came from its PCL component, while differential scanning calorimetry results confirmed that Tact was at 60 °C. The mechanical performance test results described the elastic behavior of the SMPU above Tact. The shape memory results showed that the change process had a great influence on the temporary shape and recovery shape of the sample. Compared with the sample that changed at 25 °C, the sample that changed at 80 °C had a larger shape fixation rate and a lower shape recovery rate.

In summary, polyurethane elastomer material in drilling fluid plugging has a certain degree of application, but, for its performance improvement, it also requires a certain degree of research.

##### Application of Elastic Graphite Plugging Material

After the oil-based drilling fluid in the shale gas production layer leaks, the conventional water-based plugging material is mainly used for plugging operations. Therefore, the material does not expand and does not form a network under high temperatures and in oil-based mud environments. The strength of the plugging layer is low, not dense, and the compressive strength is not enough to meet the needs of plugging. The success rate of one-time plugging is low, and the leakage is repeated. Water-based plugging materials such as walnut shell, sawdust, while drilling, plant fiber, etc., swell in water-based drilling fluid, bridge and mesh with each other, have high plugging efficiency and strong pressure bearing capacity. Under oil-based conditions, it shows an oil-repellent state and is incompatible with oil-based drilling fluid, so it cannot form a network structure and effectively seal the leakage layer. If the formation fracture expands due to the increase in pressure difference, the ordinary plugging material cannot meet the plugging demand, and the elastic graphite carbon will expand, which can still maintain the good sealing of the wellbore and prevent the leakage of drilling fluid, which is beneficial to reduce the drilling cost.

Wang et al. [81] developed the wellbore pressure-bearing plugging agent PF-RG. This plugging agent has excellent properties such as complete inertia, non-magnetic and high temperature resistance. It has good adaptability in water-based drilling fluids and oil-based drilling fluids. It does not affect the rheology, reduces the lubrication coefficient and greatly reduces the filtration loss. After evaluation of the compression elasticity experiment, the material has very high rebound characteristics, a rebound rate of more than 20%, up to more than 100%, comparable to similar foreign products. At present, the wellbore pressure sealing agent PF-RG has been successfully used in field operations, which proves that the material has excellent pressure sealing ability.

#### 3.1.2. Plugging Performance Test of Elastomer Plugging Material

Hu [82] formed a modified graphite plugging agent by continuously forging petroleum coke at 2400 °C. It has better elastic properties due to its high carbon content of 99.5% and a more ordered crystal structure. Experiments show that the material, at up to 425 °C: temperature conditions, in mineral oil or crude oil, there will be no softening phenomenon, more than 80% of which is expanded graphite and flake graphite, the crystal has a corner, edge, surface, free corner atoms, the metal surface and the wellbore has strong adhesion, can effectively reduce the friction between the drilling tool and the wellbore, reduce the construction friction [83].

The evaluation of the plugging effect of a modified graphite is as follows. Through visual sand bed and high-pressure sand bed plugging experiments, it is found that adding 3% modified graphite plugging agent into OBDFs can plug an artificial sand bed prepared by suppressing sands of 20~40 meshes at ambient temperature and pressure, with small invasion depth and no GTX-3 leakage. When the temperature reaches 120 °C, it can also seal the sand bed, and the pressure bearing capacity reaches 7 MPa, showing good lipophilic ability and temperature resistance sealing ability. When it comes to fracture plugging experiments (Table 9), it has been shown that the addition of 3% modified graphite plugging agent OBDFs can effectively plug 1 mm cracks, while compounding 5% asphalt resin plugging agent can effectively seal 2 mm cracks, and reach a bearing capacity of 7 MPa. It was found that, after adding conventional water-based plugging materials to OBDFs, the friction coefficient of mud cake was increased, and large particles produced faster friction coefficient increase. When GTX-3 was added, the friction change was relatively small and the bearing strength was high, which is especially suitable for the plugging operations of OBDFs.

#### 3.1.3. Conclusion and Prospect of Elastomer Drilling Fluid Plugging Agent

The downhole crosslinked elastomer can reduce the thickening of cement slurry by groundwater and cement slurry, and improve the success rate of water plugging. At the same time, the characteristics of downhole crosslinking can cut off the backflow and transport the material to a predetermined position that is more controllable, which is a major direction for the development of drilling fluid plugging agent in the future. However, downhole crosslinking means that the required plugging elastomer is crosslinked in a relatively complex downhole environment. Such a complex environment poses a challenge to subsequent research, meaning that the research direction must migrate to optimizing crosslinking conditions. At the same time, the more complex and more extreme downhole environment also puts forward more stringent requirements for the living environment of the plugging agent obtained by downhole crosslinking. Therefore, it is also a major problem in the future to study the performance of downhole crosslinked elastomer plugging materials.

Polyurethane elastomer material has excellent physical properties, wide adjustable range, can meet the needs of different environments and has good tear strength and tensile strength. The biggest advantage of this material is the self-healing ability brought by its shape memory function, which can be very suitable for drilling fluid plugging agents. However, correspondingly, the disadvantages of this kind of elastomer material are also obvious. Its shape memory function is greatly restricted by the external environment. The shape memory polyurethane elastomer material studied at present can show good performance below 100 °C. Above this temperature, the performance will be greatly reduced, and the drilling fluid requires a higher temperature, so this will be a great problem for future research.

The advantage of elastic graphite material lies in its expansibility. In addition, based on the resilience of expansibility, it has better plugging performance than walnut shell, sawdust and other substances. It is a plugging material worthy of study. However, the characteristics of graphite not resistant to high temperature are the problems that it needs to be urgently overcome. Therefore, most of the time, graphite-based materials still appear as composite materials, which can be used as elastomer reinforced materials to improve performance.

In summary, the elastomer material has excellent basic properties, and there has been some research progress in the field of plugging. As a plugging material, it has a good development prospect.

### 3.2. Elastomer Lubricating Material

High friction and high torque are one of the most difficult problems for engineers in long extended wells and long horizontal wells. Drilling fluid additives are the most effective way to solve this problem. Lubricant additives are added to drilling fluids to lower the drag and torque between the drill strings and rock formation. They impart lubricating properties into two moving surface contacts under extreme temperature and pressure conditions [84]. Generally, the friction coefficient of oil-based drilling fluid is about 0.08, while that of water-based drilling fluid is more than 0.2, which is much higher than that of oil-based drilling fluid [85].

With the continuous development of horizontal wells, highly deviated wells, deep wells and extended reach wells technology, as well as increasing environmental requirements, the variety of lubricants is being quickly and continuously updated. In the past decade, solid lubricants have developed the fastest, represented by plastic balls, glass beads and graphite. Studies have shown that the wear of casing metal is related to the energy consumed by friction during wear. High-quality lubricants in drilling fluids may reduce the friction coefficient, but it is impossible to reduce wear. The solid lubricant can produce physical separation between the two contact surfaces, while the liquid cannot.

#### 3.2.1. Application of Elastomer Lubricating Materials

##### Application of Elastic Graphite Lubricating Material

Bryant et al. [86] studied the friction and wear mechanisms of graphite. Elastic graphite comprised of mainly expandable graphite and flake graphite. It has strong adhesion to the metal surface and the wellbore. Therefore, it can smooth the wellbore and can form a scale film through the formation of wellbore protective film, transforming the sliding friction between the metal surface of the drill pipe and casing into a sliding friction between graphite flakes, thereby reducing the drilling fluid friction, and also reducing the wear between the drill pipe and casing surface. After the elastic graphite is pumped into the well, the particles will enter the formation pores or micro-cracks under differential pressure, where elastic deformation occurs. Laboratory tests have shown that the deformation of elastic graphite helps to increase the strength of the formation, thereby increasing the pressure that causes the formation to rupture. The graphite volume in each elastic graphite carbon particle accounts for more than 50%, which facilitates it to lubricate and make it easier to move and fill into the cracks.

Zaidi et al. [87] proposed that lamellar materials as graphite are used in several technological fields. Because of its anisotropic properties, graphite is a good material of friction. Its covalent properties prevent jamming in a sliding contact and its lamellar aspect reduces shear strain.

Faleh et al. [88] found that in order to further improve some properties of composite materials, especially friction properties, several solid lubricants were used in the production process of composite materials, among which elastic graphite was particularly prominent in the field of lubrication.

Elastic graphite has good compatibility with WBDFs, has little effect on rheology, can reduce the filtration loss of drilling fluid and can significantly improve the lubrication performance of drilling fluid, and exhibits excellent temperature resistance.

##### Application of Epoxy Elastomer Lubricating Material

Hamed et al. [89] used surface-initiated atom transfer radical polymerization (Si-ATRP) on the hydrophilic polydimethylsiloxane (PDMS) elastomer surface of polyacrylic acid (PAAc). The obtained hydrophilic PDMS was stable in air and had film lubrication behavior under water conditions.

Jia et al. [90] successfully synthesized self-lubricating microcapsules with liquid paraffin as the core material using polyurethane (PU)/silica (SiO_2_). They used self-lubricating microcapsules as a filling matrix and compounded it with PU elastomers by in situ polymerization. Compared with pure PU elastomer, the friction coefficient and wear of the composites were reduced by 64.8% and 66.7%, respectively. At the same time, PU composites showed better thermal stability and mechanical properties. The thermal decomposition temperature, tensile strength and elongation at break were increased by 4.6%, 69.1% and 30.8%, respectively, which extended the service life of the PU elastomer in special fields.

After a long period of experimental research, it was found that the epoxy elastomer material has certain lubricating properties and relatively excellent physical properties, and has potential as a drilling fluid lubricant.

##### Application of Polyurethane Elastomer Lubricating Material

Mao et al. [91] developed a new potential method for the large-scale preparation of self-lubricating TPU composites based on melt processing of TPU with polyurea formaldehyde (PUF) microcapsules containing lubricating base oil. Compared with pure TPU, the introduction of 10 wt.% microcapsules in TPU significantly reduces the friction coefficient by 78.8%, and improves its tribological properties without sacrificing or even enhancing its mechanical properties.

#### 3.2.2. Lubrication Performance Test of Elastomer Lubricating Material

After adding elastic graphite into the base mud at different dosages, the rheology, filtration and lubrication performance of drilling fluid were tested to determine the lubrication effect and reasonable dosage of elastic graphite. The base slurry formula was: 4% bentonite + 0.5% NaCO_3_ + 0.1% XY-27 + 0.3% FA367 + 0.2% CMC(MV) + 3%SMP-1 + 3% SPNH + 3%QCX-1. A G8 extreme pressure lubrication instrument (its friction was steel/steel) was employed in the lubricity test of drilling fluid. Two lubrication performance parameters, i.e., the lubrication coefficient of drilling fluid and the filter cake viscosity coefficient, were determined. The experimental results are shown in Table 10; the lubricity of drilling fluid was significantly improved after adding elastic graphite. The lubrication coefficient decreased from 0.28 to 0.16, and the friction coefficient of the filter cake decreased from 0.0875 to 0.0612, indicating that the elastic graphite was a good lubricant. The lubricating coefficient of the drilling fluid shows a trend of first decreasing and then increasing with the increase in graphite content. When the amount of elastic graphite was 0.3%, the lubrication coefficient of the drilling fluid and the viscosity coefficient of the filter cake reached the lowest point. Combined with the experimental results of drilling fluid rheology and filtration, it can be determined that when the amount of elastic graphite was added at 0.3%, the rheology and lubrication performance of the drilling fluid simultaneously reached a superior level.

It can be seen from Table 10 that the improvement of the lubrication coefficient of the drilling fluid by elastic graphite was more significant. After aging, the lubrication coefficient of base slurry could be reduced by 30.4% and the lubrication effect was better than that at room temperature. While several other lubricants had obvious deficiencies in improving the lubrication coefficient of drilling fluid. It was shown that the lubrication effect was worse after aging at 150 °C, than that at room temperature. On the other hand, from the perspective of the reduction rate of the viscosity coefficient of the filter cake, the effects of several lubricants are comparable. Although the effect of elastic graphite was not the best, it could also reduce the viscosity coefficient of the filter cake of the drilling fluid by about 30%, which greatly improved the lubricity of the filter cake.

#### 3.2.3. Conclusion and Prospect of Elastomer Drilling Fluid Lubricant

Elastic graphite is mainly composed of expandable graphite and flake graphite. It has a strong adhesion to the metal surface and the borehole. It can convert the sliding friction between the metal surface of the drill pipe and the casing into the sliding friction between the graphite sheets, thereby reducing the friction of the drilling fluid and reducing the wear between the drill pipe and the casing surface. However, the biggest problem of elastic graphite is that its own temperature resistance is too poor to be used as a lubricant in extreme drilling environments. In future research, researchers can focus on the mechanism of elastic graphite lubrication to see whether it can be found. The effectiveness of elastomer materials with similar mechanisms to replace elastic graphite, or directly use it as a polymer material to work with other materials without losing the corresponding lubrication effect.

Epoxy elastomer can form a lubricating film in the presence of water to reduce friction. The corresponding lubrication effect is determined by the friction size of the lubricating film. Therefore, future research on the lubrication performance of this elastomer material should focus on the chemical composition and type of the lubricating film, and strive to improve the lubrication performance.

#### 3.2.4. Application of Elastomer Fluid Loss Material

Jain and Mahto [92] evaluated the feasibility of polyacrylamide-grafted-polyethylene glycol/silica and polyacrylamide/clay nanocomposites as potential additives for the drilling fluids.

Mohamadian et al. [93] used microemulsion polymerization to prepare styrene-methyl methacrylate-acrylic acid/nanoclay hybrid polymer nanocomposites as drilling fluid additives. The polymer/clay composite nanoparticles significantly improved the rheology and filtration performance of the drilling fluid and remained stable under high pressure, high temperature and harsh salinity conditions.

#### 3.2.5. Test of Fluid Loss Reduction Performance of Elastomer Fluid Loss Reduction Material

After adding elastic graphite into the base mud according to different additions, the rheology, filtration and lubrication performance of the drilling fluid were tested to determine the lubrication effect and reasonable amount of elastic graphite. The base slurry formula was: 4% bentonite + 0.5% NaCO_3_ + 0.1% XY-27 + 0.3% FA367 + 0.2% CMC(MV) + 3%SMP-1 + 3% SPNH + 3%QCX-1.

The elastic graphite was added to the base slurry at 0.1%, 0.2%, 0.3%, 0.4% and 0.5%, respectively. The rheology and filtration of drilling fluids with different graphite additions were compared, and the results are shown in Table 11. The effect of elastic graphite on drilling fluid rheology and filtration was as follows.

It can be seen from Table 11 that the rheology of drilling fluid did not significantly change after adding elastic graphite, indicating that elastic graphite did not affect the rheology of drilling fluid. Some other lubricant products will produce dilution, so that the drilling fluid shear force is reduced, which can be seen in the later elastic graphite and other lubricant effect comparison experiment. At the same time, the filtration loss of the drilling fluid significantly decreased, from 4.7 mL to about 3.0 mL. The reason is that the elastic graphite has flexibility and elasticity, so that when the drilling fluid filtration experiment is carried out, the graphite particles enter the pores of the filter cake under the action of pressure difference, blocking a part of the larger pores, thus reducing the water loss of the drilling fluid.

#### 3.2.6. Conclusion and Prospect of Elastomer Drilling Fluid Filtrate Reducer

The study of polymers used in drilling fluid loss reduction is very common. From the performance test of the corresponding elastomer material drilling fluid, it can be seen that the elastomer material will also play a certain role in the field of drilling fluid loss reduction. In future research, this is a topic worthy of discussion. Whether the characteristics of the elastomer itself have a certain impact on the loss reduction, or whether the elastomer and other polymers will enhance the processing performance of the corresponding loss reduction material.

At present, the application of elastomer materials as fluid loss reducers has not been reported. Existing experimental studies have shown that the corresponding fluid loss reduction effect is based on certain elastomer plugging characteristics, and subsequent development and research are needed.

## 4. Preparation of Elastomer Powders

In order to apply the elastomer material to the drilling fluid, elastomer material needs to be crushed, so that the elastomer material can be used as an additive for drilling fluid to play the role of lubrication and plugging. This section summarizes the crushing methods of elastomer materials, mainly focusing on the preparation methods of powder rubber.

### 4.1. Development of Powder Rubber

Generally, powder rubber refers to rubber with an average particle size of less than 1 mm. Powder rubber has significant advantages over traditional block rubber.

Powder rubber can be processed by traditional methods or plastic processing machinery. The former can save energy consumption and shorten processing time. The latter can synthesize mixing and forming into a process, which is continuously completed by an extruder; at the same time, the continuous measurement, mixing, molding and vulcanization can be realized. Applications: powder rubber can not only be prepared rubber products, but also as a modifier, widely used in the field of adhesives and polymers. The powder rubber particles can be modified by core or shell to obtain special functions and have a wider application field.

The first technical paper on powdered rubber was published by Dunlop in 1930. In 1956, the United States BFGoodrich company first produced powdered nitrile rubber, commodity brand Hycar-1411. Since then, many foreign manufacturers and research institutions have actively carried out the development and research of powder rubber. After the low tide period in the 1960s and 1970s, due to the requirements of the tire processing industry, the rapid development of the polymer modification industry and to meet the requirements of environmental protection and promote the use of waste rubber, powder rubber technology saw rapid development; at the same time, as powder rubber production and processing technology progressed, powder rubber research began in earnest. At this time, powder rubber technology was mainly based on physical methods such as mechanical crushing methods and spray drying methods, and chemical methods were represented by the coagulation method. Due to the high production costs of the mechanical crushing methods and spray drying methods, the product quality was not satisfactory, and the variety of powder rubber produced was limited. Therefore, the most direct and economical powder rubber preparation and condensation methods began to gradually occupy the dominant position of powder rubber technology in the late 1990 s.

In the 1970s, the domestic research of powdered rubber began. The preparation of powdered nitrile rubber by spray drying technology was first studied by Lanzhou Chemical Industry Company Synthetic Rubber Plant. In 1987, the factory adopted the technical route of preparing powder nitrile rubber by latex direct coacervation, and carried out a significant amount of experiments. The plant pioneered the condensation method by the system of automatically generated isolation agent technology, which has two patents. In February 1991, the technology passed the small-scale test organized by the China Petrochemical Corporation. Its production of powdered nitrile rubber for the manufacture of friction materials and PVC-modified industry could replace the United States BFGoodrich company Hycar-1411; in 2001, Lanzhou Petrochemical Research Institute developed the technology of preparing various kinds of powdered rubber using the self-generated isolating agent agglomeration method, which has broad prospects. In 2001, the Beijing Research Institute of Chemical Industry of Sinopec successfully developed the radiation crosslinking coacervation method and built a 500 t powder rubber production plant. Radiation crosslinking condensation technology is the first in China and abroad, which represents a new technical direction in the field of powder rubber. The powder rubber prepared by this technology has the advantages of small average particle size (20~500 nm) and many kinds of rubber, and has a very wide range of uses in the field of polymer modification. The method can produce nanoscale fully crosslinked powder nitrile rubber, powder styrene butadiene rubber, powder chloroprene rubber and powder acrylate rubber. In 2002, Lanzhou Chemical Industry Co., Ltd. Latex Center built a powder rubber production plant with an annual output of 2000 tons using the technology of the Lanzhou Petrochemical Research Institute.

### 4.2. Powder Rubber Production Technology

The production of rubber powder is based on four major raw materials, namely block rubber (raw rubber, vulcanized rubber), latex, rubber liquid and suspension, and two major technologies, namely powdering technology and isolation technology. Many powdered rubber production methods have been derived from four raw materials and two technologies.

At present, there are many technologies for the preparation of powdered rubber, which can be classified according to different properties. According to the classification of raw materials, it can be divided into solid rubber, such as the crushing method; using latex and rubber suspension, such as the drying method, the coagulation method, the microcapsule method, etc. According to the characteristics of the technology itself, it can be divided into physical methods and chemical methods. Physical methods include the crushing method, the drying method, etc. Chemical methods include the agglomeration method, the microcapsule method, etc.

Today, raw rubber powder is mainly made using the spray drying method and the coagulation method, and the waste tire powder is made using the crushing method. Therefore, these three mature and representative technologies are introduced below.

#### 4.2.1. Crushing Method

Crushing is the earliest technology used in the field of powdered rubber, mainly used for solid rubber, including waste rubber products and tires. Therefore, this technology has an important position in the reclaimed rubber industry. Common crushing methods include the frozen crushing method and the normal temperature crushing method.

##### Frozen Crushing Method

The frozen crushing method is used to deal with block and granular raw rubber, waste rubber products, waste tires and other solid rubber. The rubber is frozen to below the glass transition temperature for crushing, and then a certain amount of isolation agent is added, then fine rubber powder with a particle size of 75~300 μm can be prepared. According to the process, the frozen crushing method can be divided into an entirely low temperature method, which includes coarse crushing and fine crushing carried out at low temperatures; a normal temperature low temperature method, which includes coarse crushing and fine crushing at normal temperatures and low temperatures, respectively. According to the freezing medium, the low temperature crushing method can be divided into liquid nitrogen low temperature crushing and air expansion refrigeration crushing. After using liquid nitrogen as a cooling medium to freeze the rubber, the frozen rubber is then crushed using a toothed disk or hammer mill. Due to the high costs of the liquid nitrogen freezing method and expensive equipment, it is not widely applied. In recent years, a technology for preparing powdered rubber has been developed by air turbine expander refrigeration, in which air expansion is employed to refrigerate and freeze rubber and then crush it by grinding or using a jet mill. From a process perspective, air turbine expander refrigeration, when crushing rubber, generally needs to go through normal temperature coarse crushing and frozen fine crushing as two steps; the liquid nitrogen freeze crushing method can be used through the above two steps, can also be crushed under full freezing. From the perspective of practical application, the air turbine expander refrigeration method has lower costs and a stronger implementation than the liquid nitrogen refrigeration method.

In the process of preparing powdered rubber by freeze crushing or after crushing, isolation agents (such as silicone oil, carbon black, silica, etc.) are added to the powdered rubber to prevent powder adhesion. The powdered rubber is either impregnated in an aqueous dispersion of the isolating agent and then dried and dehydrated to ensure the fluidity of the powdered rubber. The rubber powder produced by the low temperature crushing method has small particle size, a smooth surface and a low degree of heat oxidation, but the production cost is high. It is mainly used in high performance products, such as radial tire tread, high performance plastics, coatings, adhesives and military products.

##### Normal Temperature Crushing Method

The normal temperature crushing method generally refers to the crushing operation at (50 ± 5) °C or a slightly higher temperature. The raw materials used are bulk unsulfured rubber or recycled vulcanizate (such as waste tires). Rubber is an elastomer at room temperature, so the rubber crushing at room temperature is different from other materials crushing. The crushing mill for crushing rubber at room temperature is still in the research and developmental stages, and its process design has not been finalized. A rotary micro-pulverizer is used to crush at room temperature, and there is also a method of mutual abrasion of rubber through forced air flow. The normal temperature crushing method is carried out through mechanical shearing, tearing, crushing and other effects of crushing, so the resulting product powder surface is uneven, burr-like. This kind of rubber powder has a larger specific surface area than the rubber powder prepared using the freezing grinding method, so it is beneficial to its activation modification. At the same time, when it is mixed with the raw rubber powder, the binding force with the raw rubber powder is large.

The normal temperature crushing method can be divided into dry and wet methods. Dry crushing, according to its crushing methods and equipment, can be divided into different roller, rotary, extrusion, high-pressure plunger types. At present, the most commonly used rotary crushing method is mainly through the use of a fixed metal knife and a rotary knife used between the shear force and the friction between rubber and the tool to complete crushing. China Jiangyin Gasification Machinery Factory developed a rotary micro—mill. Wet crushing is used with chemical agents or water to pretreat the coarsely crushed rubber, and the pretreated rubber powder is ground with a colloid mill or a grinding disk, followed by post-processing, such as chemical treatment agent removal, dehydration and drying to obtain fine rubber powder. Wet crushing creates particles that are small in size, generally more than 200 mesh, has less thermal degradation and its powder performance is better than dry crushing. The room temperature immersion mixing crushing method developed by Wuxi Boda Rubber Powder Technology Co., Ltd. is a kind of wet crushing. It has low costs, low investment, excellent product quality and high competitiveness.

Through the analysis and comparison of the technical and economic indicators of the normal temperature crushing method and the low temperature crushing method, as shown in Table 12, it can be predicted that the normal temperature crushing technology will dominate the production of rubber powder with rubber blocks, especially with waste tires. This is because, from the technical and economic indicators, low temperature crushing cannot be compared with normal temperature crushing. In particular, the normal temperature immersion mixing grinding method is suitable for China’s national conditions of waste rubber powder method.

#### 4.2.2. Agglomerative Method

The coagulation method is the most direct and economical method for preparing powdered rubber, which is the dominant technology in the field of powdered rubber technology. Its simple process, low costs, small particle size and good performance represent the development direction of powdered rubber technology. People have devoted research to the study of powdered rubber by coagulation from the beginning. However, the condensation method has three technical difficulties: first, particle size control; second, anti-sticking isolation; third, slurry filter cake re-pulverization. These were basically solved after decades of research.

The coagulation method involves adding a coagulant to the latex, so the latex is in a coarse rubber suspension. The coagulated rubber particles are isolated, dehydrated and dried to obtain powder rubber. When selecting coagulants, the coagulation ability, consumption, price and cost of the selected coagulants should be considered first. Coagulants include various inorganic salts and organic acids, polymer flocculants, such as water-soluble calcium salts, magnesium salts, aluminum salts, bone glue, etc.

The coagulation method can be divided into: 1. A self-generated isolation agent coagulation method. This method is the most advanced coagulation method in the world at present. Its advantage is that there is no need to add isolation agent in the coagulation process, and the soap in the latex is automatically converted into an isolation agent in the coagulation process. 2. An additional isolation agent coagulation method. 3. The coagulation method is coagulation under the conditions of a large number of fillers in the latex. 4. A particle surface treatment (sulfuric acid treatment, chlorination, surface crosslinking, etc.) coagulation method. The following introduces the main self-isolation agent method and co-condensation method in the condensation method.

##### Self—Generated Isolation Agent Coagulation Method

The self-isolation agent condensation method is the most advanced condensation method in the world at present. Its advantage is that no additional isolation agent is required in the condensation process. The soap in the latex is automatically converted into an isolating agent during coagulation. Therefore, the process is simplified, the consumption and cost are reduced and environmental pollution is reduced.

Lanzhou Petrochemical Research Institute has studied the direct condensation of latex into powder, and successfully developed the latex self-isolation condensation technology in 1990. The particle size of the powder obtained by this technology can reach less than 0.3 mm, and the powder rate can reach more than 90%. Only a small amount of isolation agent produced in the condensation process can achieve a good isolation effect. This technology has reached the international leading level for rubber powder production using the coagulation method. The specific preparation method is as follows: add monovalent and divalent metal salt aqueous solution as a coagulant in nitrile latex, such as low hardness salt, high hardness salt and magnesium sulfate (calcium chloride) aqueous solution, at 30~65 °C temperature pre-coagulation and flocculate, thus condense into particles. In this process, the emulsifier used in the nitrile polymerization reaction, such as potassium stearate, potassium oleate, synthetic fatty acid potassium, etc., reacts with the calcium and magnesium ions in the coagulant at 50~65 °C to form magnesium stearate (calcium), magnesium oleate (calcium) or synthetic fatty acid magnesium (calcium) and other precipitates. Because these precipitates are in a molecular state, the colloidal particles formed by latex condensation have a strong adsorption activity for them, so they have the effect of active isolation and become self-generated isolation agents to achieve anti-adhesion isolation. Because there is no need to add the usual isolation agent, the rubber powder obtained by this technology has high rubber content and good performance. At present, the rubber powder produced by this technology is mainly nitrile powder rubber products.

##### Co-Agglomeration Method

The co-coagulation method is to mix latex with a large number of fillers, and then use alkali metal salt solution and alkaline earth metal salt solution as coagulants to condense into powder. The fillers used in the co-coagulation method include inorganic fillers, organic fillers, resin emulsions, inorganic colloids, etc. On the one hand, the filler plays an isolation role to prevent the congealed particles from adhering to each other; on the other hand, it is also a component of the compound formulation. The amount of filler is generally more, about 30~70%. Inorganic fillers and organic fillers cannot be directly added to the latex; they need to be made into a stable suspension and then evenly mixed with the latex. The co-agglomeration method rubber powder has reduced the processing procedure and reduced the production cost because some components have been evenly added to the rubber powder in the processing formula. The carbon black co-precipitation method is a representative technology of the co-coagulation method. The latex is mixed with the water suspension dispersion of carbon black and then added with coagulants, such as NaCl, CaCl_2_, MgCl_2_, MgSO_4_, H_2_SO_4_, HCl, etc., so that it is co-coagulated, filtered, washed, dehydrated and dried to obtain powdered rubber. Carbon black is both a reinforcing agent and an isolation agent, and is conducive to processing, transportation and storage. In addition, in order to improve the fluidity of the obtained powder, a polymer coating agent can be further added to cover the surface of the aggregates composed of rubber and carbon black to achieve a good isolation effect. It has been reported that the polymer coating agent can be obtained by free radical copolymerization of two vinyl monomers. Powder rubber prepared using the carbon black co-precipitation method commonly includes natural powder rubber, nitrile powder rubber, SBR powder rubber, etc.

#### 4.2.3. Spray Drying Process

Spray drying is one of the most important methods for preparing powdered rubber, and it is also one of the industrialized methods for preparing powdered rubber. Almost all kinds of rubber can be made into powder by spray drying. The spray drying method is to add anti-adhesive latex, glue or suspension spray into the drying tower with a variety of atomizers, atomized into small droplets on contact with hot air and rapid drying to obtain solid powder. The spray drying process can be divided into four parts: liquid atomization; droplet contact with air; droplet drying dehydration; the dried powder is separated from air. The key equipment of the spray drying system is an atomizer, a drying tower and a separator. According to the different atomization methods, the spray drying method can be divided into three kinds: airflow spray, pressure spray and centrifugal spray. According to different raw materials into powder open, closed cycle, semi-closed cycle 3 kinds of spray system.

The spray drying method can be used for a variety of products, including latex, glue powder and anti-adhesive uses for silica-type reinforcing agents. The powder rubber prepared using the spray drying method has fine particles, high yield and low costs. However, the particle size distribution of powdered rubber prepared by this method is wide and not suitable for some applications. In order to produce powder rubber with roughly uniform particle sizes, grinding and sieving are also needed. Powder rubber prepared using the spray drying method has porous particles, which can be made into particles with uniform particle size distribution by hammer grinding and sieving. In order to prevent bonding, some isolation agents must be added to prevent re-bonding during storage and transportation.

### 4.3. Development of Powder Rubber Production Technology

In order to apply the elastomer material to the drilling fluid, the elastomer material needs to be crushed so that the elastomer material can be mixed with the drilling fluid additive in the form of powder to play the role of lubrication and plugging. However, at present, research on elastomer materials has entered a new field. Many modified polymer elastomer materials have emerged one after another. Their physical and chemical properties are far superior to those of conventional rubber materials. If they need to be crushed and used in combination with drilling fluid, the crushing technology of elastomer materials will be another major problem to be overcome. It is necessary to study new crushing methods based on the crushing of conventional rubber materials.

## 5. Conclusions


(1)The elastic body is a material that quickly recovers to its approximate initial shape and size after a large deformation occurs and the external force is withdrawn. There are many different types of elastomer materials, such as supramolecular elastomers based on hydrogen bonding, elastic graphite, fluor elastomer materials and polymer elastomers. Elastic graphite is an elastomer material which has been widely used in drilling fluids.(2)Elastomer materials can adapt to a variety of complex formation environments. Different elastomer materials have great performance and have broad application prospects in the field of drilling fluids. The elastomers used as lubricating materials can greatly change the lubrication performance of drilling fluids. The elastic material of inhibitors can improve the shale recovery rate of drilling fluids and has good ability to inhibit and prevent collapse. When used as a plugging agent, the change in drilling fluid friction is relatively small, and the pressure bearing strength is high, which is especially suitable for oil-based drilling fluid plugging operations.(3)At present, the research and application of elastomer materials in the field of drilling fluids is still in the stage of development. In the future, we should continue to improve the performance of elastomer materials, broaden the scope of application, accelerate the transformation of achievements and industrial application and promote the development of drilling fluid technology.(4)Drilling fluid applications of elastomer materials require crushing treatments of elastomer materials. In the future, we should focus on how to crush and how to combine them to promote the development of drilling fluids.


## Figures and Tables

**Figure 1 polymers-15-00918-f001:**
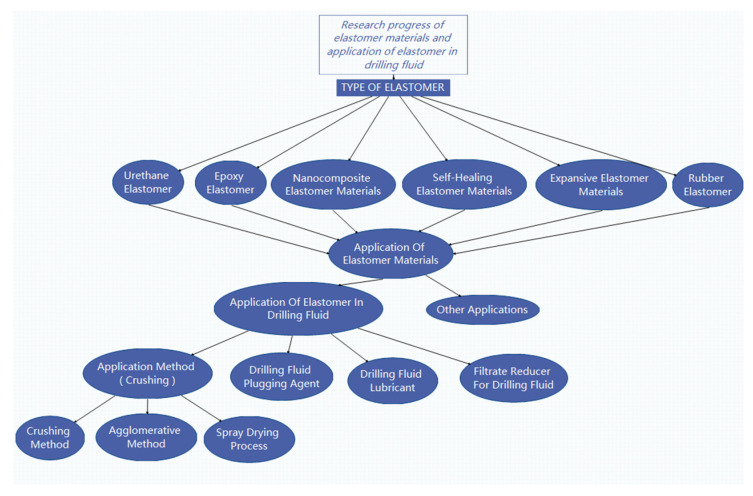
The research progress of elastomer materials and their applications in drilling fluids.

**Figure 2 polymers-15-00918-f002:**
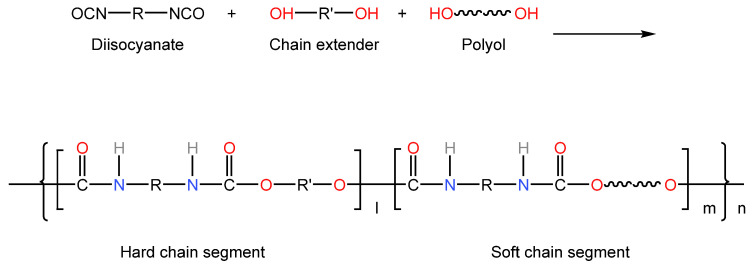
Reaction to form amino ester group.

**Figure 3 polymers-15-00918-f003:**
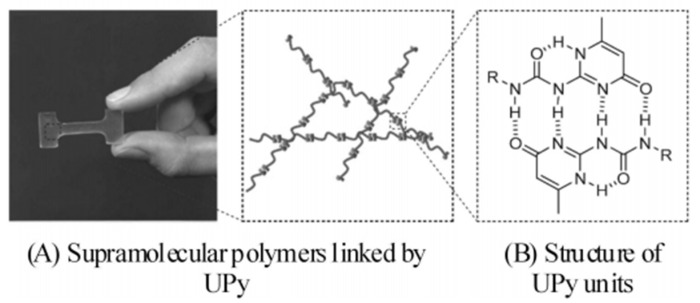
Zooming in on supramolecular polymers linked by UPy. (Reprinted (adapted) with permission from [41]. Copyright [41] American Chemical Society).

**Figure 4 polymers-15-00918-f004:**
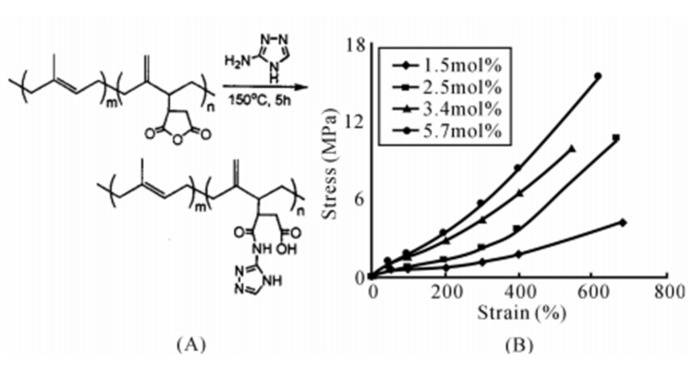
(**A**) Synthesis of IR-ATA and (**B**) stress-strain curves of IR-ATA with various graft ratios of ATA. (Reprinted (adapted) with permission from [42]. Copyright [42] American Chemical Society.).

**Figure 5 polymers-15-00918-f005:**
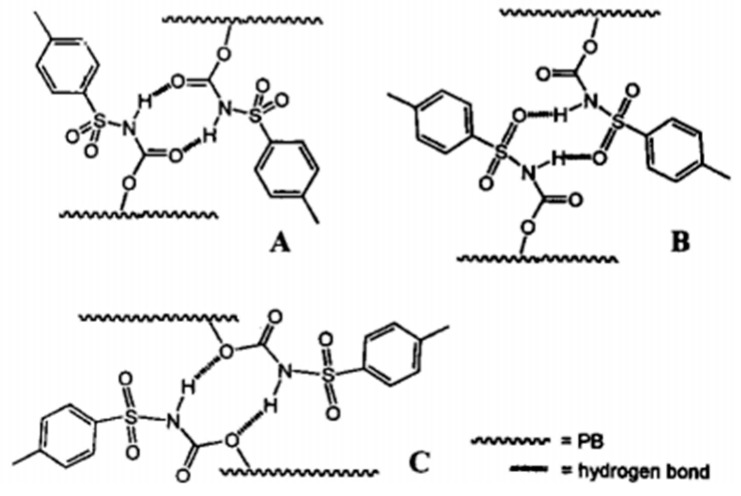
Schematic representation of three possible hydrogen-bonding complexes between two sulfonyl urethane groups (SU) grafted to the polybutadiene chains (PB). (Reprinted (adapted) with permission from [43]. Copyright [43] American Chemical Society).

**Figure 6 polymers-15-00918-f006:**
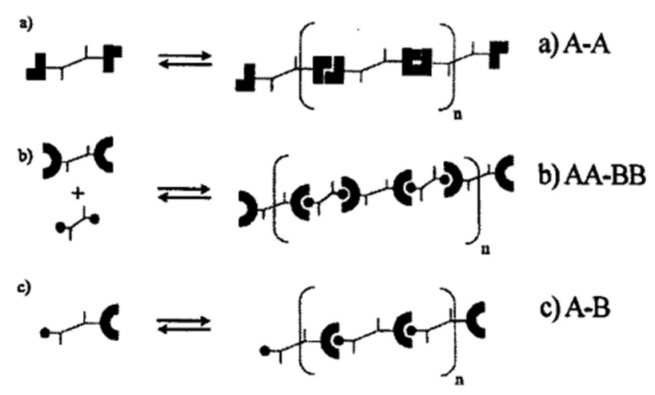
Schematic representation of supramolecular polymers assembled from self-complementary.

**Figure 7 polymers-15-00918-f007:**
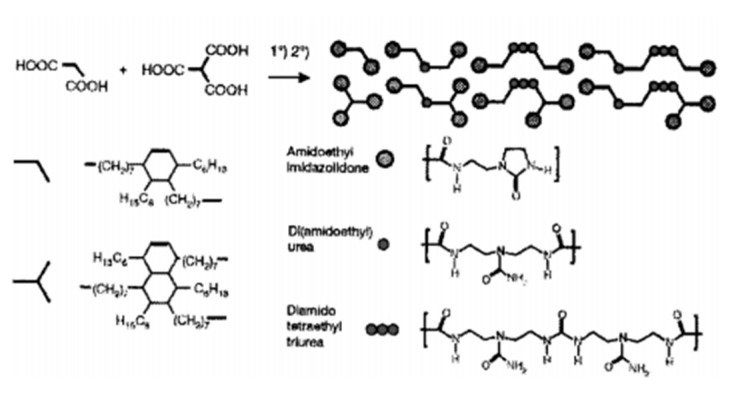
Schematic illustration and molecular structures of the self-healing rubber materials based on fatty D-I Acid and Tr-I Acid.

**Figure 8 polymers-15-00918-f008:**
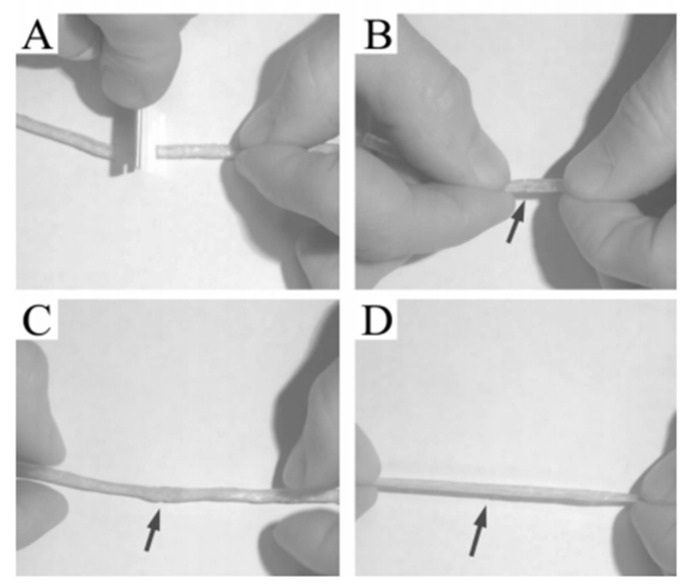
The self-healing testing of supramolecular elastomer (**A**): Cut; (**B**): Join; (**C**): Mend; (**D**): Stretch.

**Figure 9 polymers-15-00918-f009:**
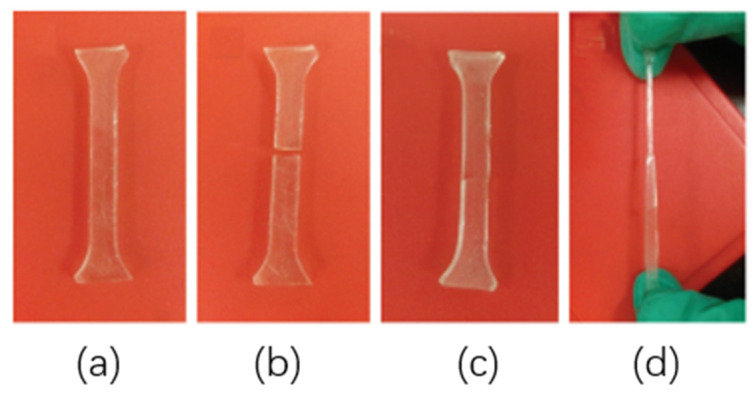
Photos of elastomer self-repair. (**a**) Original sample (**b**) After cutting off (**c**) After self-healing (**d**) Artificial constant pulling.

**Figure 10 polymers-15-00918-f010:**
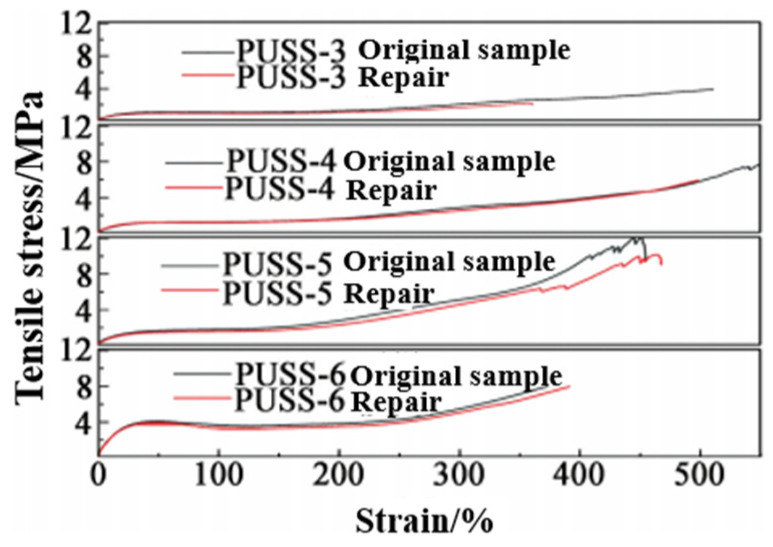
Tensile stress-strain curves of samples with different disulfide contents.

**Figure 11 polymers-15-00918-f011:**
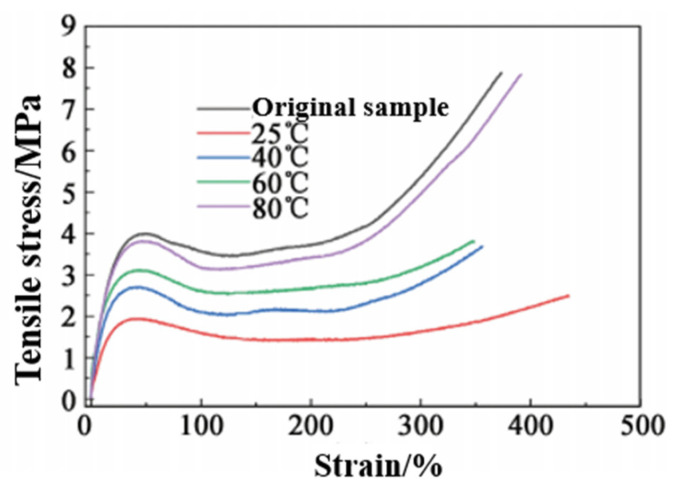
Tensile stress-strain curve of PUSS-6 after self-repair at different temperatures.

**Figure 12 polymers-15-00918-f012:**
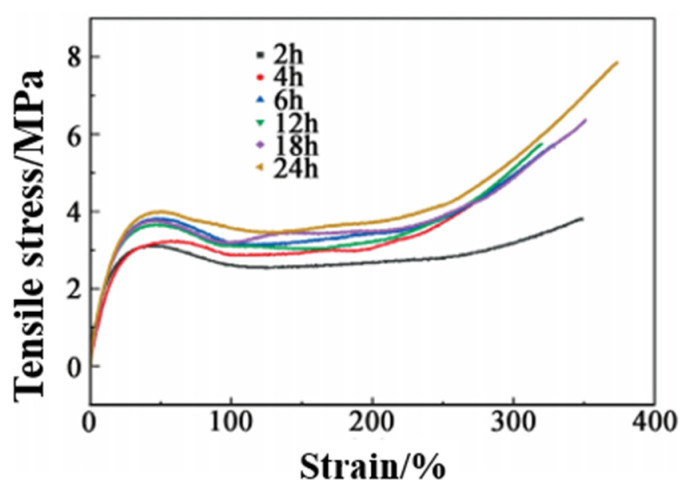
Tensile stress-strain curves of PUSS-6 after self-repair at different times.

**Figure 13 polymers-15-00918-f013:**
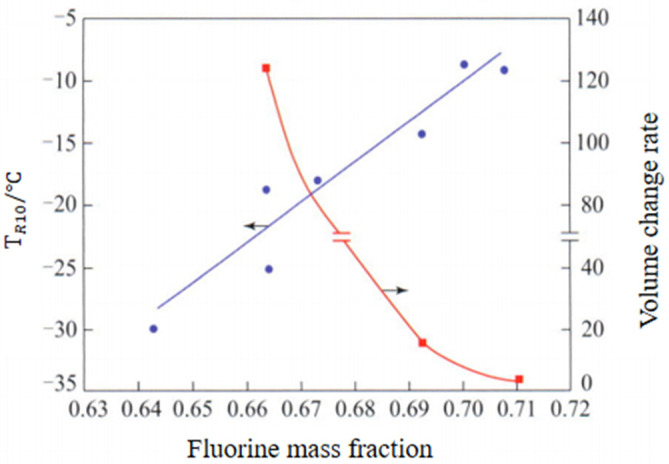
Effect of fluorine contents on low temperature resistance and methanol resistance of nuoroelastomers.

**Figure 14 polymers-15-00918-f014:**
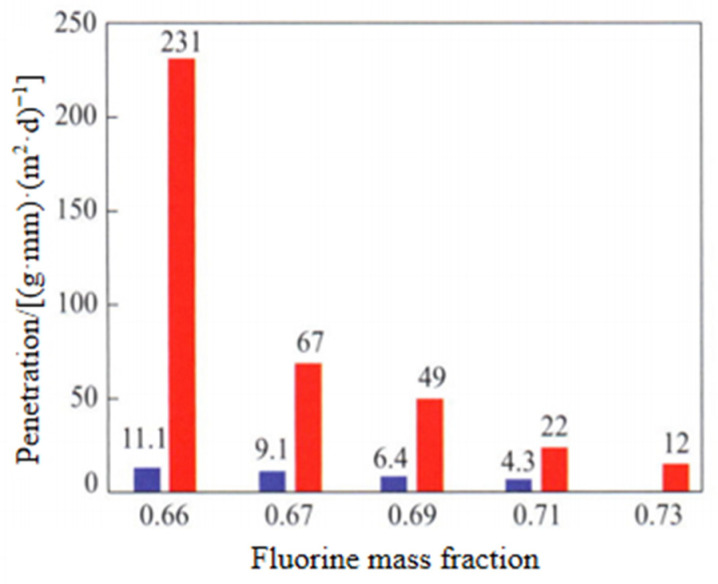
Effect of fluorine contents on fuel permeation rates of fluor elastomers. Blue column: fuel oil C; Red column: CM20 (fuel oil C/methanol is 80/20).

**Figure 15 polymers-15-00918-f015:**
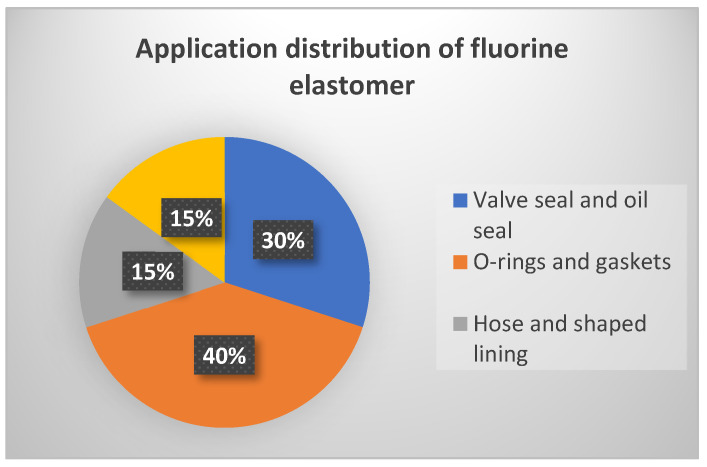
Distribution of applications of fluorine elastomers.

**Figure 16 polymers-15-00918-f016:**
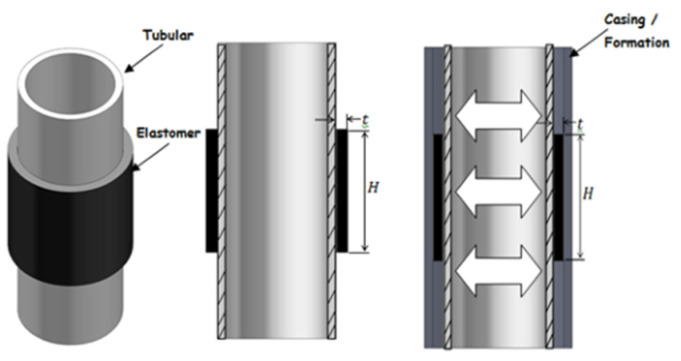
Schematic diagram of a typical expandable packer.

**Figure 17 polymers-15-00918-f017:**
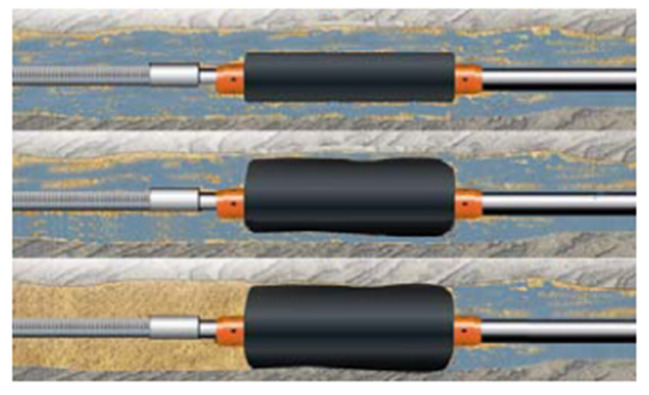
Elastomer swelling creates interzone isolation.

**Figure 18 polymers-15-00918-f018:**
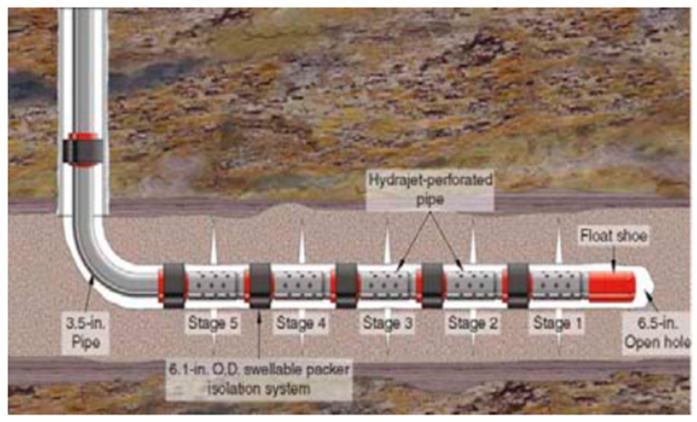
Hydra-jet perforation system.

**Figure 19 polymers-15-00918-f019:**
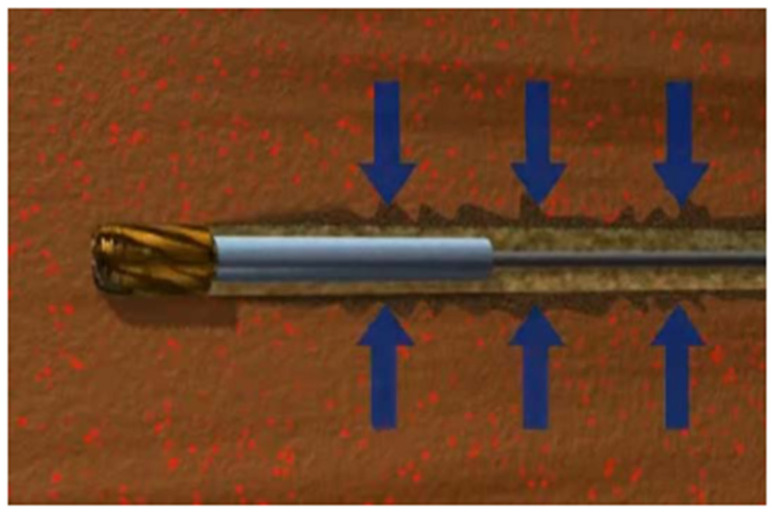
Underbalanced drilling.

**Figure 20 polymers-15-00918-f020:**
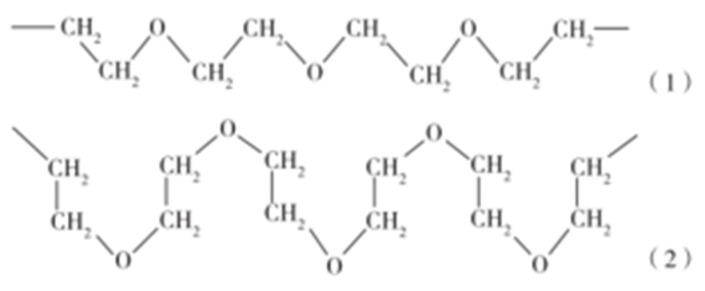
Zigzag and tortuosity molecular structures of polyethylene oxide. (1) zigzag; (2) tortuosity [70].

**Figure 21 polymers-15-00918-f021:**
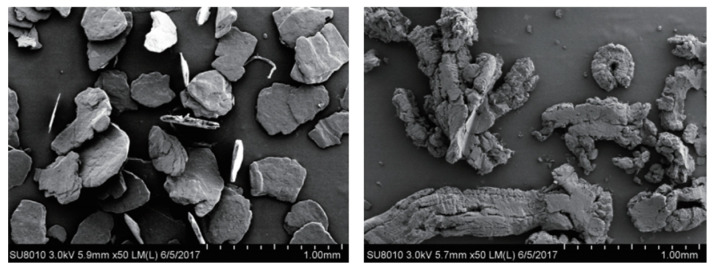
SEM images of flake graphite (**left**) and expanded graphite (**right**) [71].

**Table 1 polymers-15-00918-t001:** Raw materials suitable for TPU elastomer synthesis.

Oligomeric Diols	Diisocyanate	Chain Extender
polybutylene adipate diol	MDI	1,4-butanediol (BDO)
poly-ε-caprolactone glycol	PPDI	1,4-bis (2-hydroxyethoxy) benzene (HQEE)
Poly(tetramethylene) glycol	1,5-Naidiisocyanate	ethylene glycol (EG)
polycarbonate diols		diethylene glycol (DEG)

**Table 2 polymers-15-00918-t002:** Tensile strength and self-repair efficiency of samples with different mass fractions of disulfide.

Specimen	W(S-S)/%	Tensile Strength/MPa		Self-Healing Efficiency/%
		Original Sample	Repaired Sample	
PUSS-3	3	3.73	2.03	54.5
PUSS-4	4	8.02	5.80	72.3
PUSS-5	5	11.90	10.05	84.5
PUSS-6	6	7.88	7.85	99.5

**Table 3 polymers-15-00918-t003:** Tensile strength and self-repair efficiency of Puss 6 after self-repair at different temperatures.

Self-Healing Temperature/°C	Tensile Strength/MPa		Self-Healing Efficiency/%
	Original Sample	Repaired Sample	
25	7.88	2.47	31.3
40		3.68	46.7
60		3.81	48.3
80		7.85	99.5

**Table 4 polymers-15-00918-t004:** Elastic modulus and yield stress of PUSS-6 after self-repair at different temperatures.

Self-Healing Temperature/°C	Elastic Modulus/MPa		Yield Stress/MPa	
	Original Sample	Repaired Sample	Original Sample	Repaired Sample
25	20.39	11.13	3.99	1.94
40		16.05		2.68
60		19.67		3.10
80		20.06		3.79

**Table 5 polymers-15-00918-t005:** Tensile strength and self-repair efficiency of PUSS-6 with different self-repair time.

Self-Healing Time/h	Tensile Strength/MPa		Self-Healing Efficiency/%
	Original Sample	Repaired Sample	
2	7.88	3.81	48.4
4		4.73	60.0
6		5.74	72.8
12		5.75	73.0
18		6.37	80.8
24		7.84	99.5

**Table 6 polymers-15-00918-t006:** Five kinds of KFM elastomer monomer composition.

Class	Component
1	HPE and VF2
2	TFE, HFE and VF2
3	TFE, VF2 and A fluoroethylene ether
4	TFE, propylene and VF2
5	TFE, HFE, VF2, ethylene and A fluoroethylene ether

**Table 7 polymers-15-00918-t007:** Partial applications of fluor elastomers.

Auto-Industry	Aerospace Industry	Other Industry
1. Shaft sealing	1. O-Seals in Fuel, Lubricating and Hydraulic Systems	1. Hydraulic O-ring
2. Tire valve seal	2. Composite liner	2. Pump-ashore units check valve ball
3. Fuel nozzle O-ring	3. Tank liner	3. Lighting cartridge adhesive
4. Fuel pipe	4. Fireproof layer seal	4. Diaphragm
5. Oil tank and oil tank connection seal	5. Engine lubricating oil pipe	5. Electric clipper connection
6. Valve gasket and compound gasket	6. Jet engine pipe clamp	6. Fuel tube connection
7. Pump-ashore units check valve ball	7. Tire stem seal	7. Valve gasket
8. Machine tool cutting gasket		8. Roller coating film

**Table 8 polymers-15-00918-t008:** Selection Guide for Fluorine Elastomer.

Demand	A-HV	GBL-S	GF-S	ETP-S	TFE/P
Mechanical performance in high temperature	++	++	+	NR	NR
Methanol-tolerant	NR	NR	+	++	+
Amine-resistant preservatives	NR	NR	NR	++	++
Format-resistant solution	NR	NR	NR	++	N/A
Lipid-resistant drilling fluid	+	N/A	NR	NR	NR
Explosion resistant decompression	+	+	+	+	N/A
Compression permanent deformation resistance	++	+	++	+	NR
Low temperature elastic properties	++	++	+	+	NR

Notes: ++: Recommended selection; +: Can be selected; NR: Not recommended; N/A: Untested.

**Table 9 polymers-15-00918-t009:** Plugging test on crack plates with different crack widths.

	Testing Instrument	Crack Opening/mm	API/mL	Bearing Capacity /MPa	Plugging Situation
1	OBDF	1	complete loss	0	Cannot
2	OBDF + 3% GTX-3	1	370	3.0	Can
3	OBDF + 3% GTX-3 + 5% Asphalt Resin Plugging Agent	1	45	7	Can
4	OBDF	2	complete loss	0	Cannot
5	OBDF + 3% GTX-3	2	complete loss	0.7	Cannot
6	OBDF + 3% GTX-3+ 5% Asphalt Resin Plugging Agent	2	160	7	Can

**Table 10 polymers-15-00918-t010:** Effects of different types of lubricants on the lubrication performance of drilling fluid.

Graphite Addition	Test Condition	lubrication Factor	Lubrication Coefficient Reduction Rate	Viscosity Coefficient of Filter Cake	Reduction Rate of Filter Cake Viscosity Coefficient
Basic mud	Before aging	0.28	-	0.0963	-
	After aging	0.23	-	0.875	-
0.3% elastic graphite	Before aging	0.21	25%	0.0524	45.6%
	After aging	0.16	30.4%	0.0612	30.1%
0.3% plastic ball	Before aging	0.23	17.9%	0.0524	45.6%
	After aging	0.25	−8.7%	0.0437	50.1%
1% RH101	Before aging	0.21	25%	0.0612	36.4%
	After aging	0.27	−3.6%	0.0524	40.1%
1% RH102	Before aging	0.21	25%	0.0524	45.6%
	After aging	0.26	−13%	0.0524	40.1%
1% DG-5B	Before aging	0.21	25%	0.0612	36.4%
	After aging	0.16	17.4%	0.0524	40.1%

**Table 11 polymers-15-00918-t011:** Effect of elastic graphite on rheology and filtration of drilling fluid.

Graphite Addition	Test Condition	AV (mPa·s)	PV (mPa·s)	YP (Pa)	G_10s_/G_10min_ (Pa/Pa)	API (mL)
0	Before aging	25.5	21	4.5	1/7	3.6
	After aging	31.5	24	7.5	1.5/7	4.7
0.1%	Before aging	28.5	17	11.5	1/7	3.9
	After aging	27.5	20	7.5	1.5/7	4.7
0.2%	Before aging	25	19	6	1/3.5	3.9
	After aging	25.5	17	8.5	1/6	4.2
0.3%	Before aging	30.5	19	11.5	1/5	3.3
	After aging	28	20	8	1.5/7	3.3
0.4%	Before aging	25	15	10	0.5/4.5	3.0
	After aging	39	30	8	5/8.5	3.5
0.5%	Before aging	31	22	9	1/5	3.0
	After aging	33	23	10	1.5/7.5	3.6

**Table 12 polymers-15-00918-t012:** Comparison of technical and economic indexes between normal temperature crushing method and low temperature crushing method.

Manufacturing Method	Normal Temperature Crushing Method	Freeze Grinding	Air Expansion Refrigeration Crushing
Ten thousand tons of product equipment investment million yuan/a	1300	4000	2000
Production cost yuan/t	1800	6000	4000
Number of employees	12	22	22
Energy consumption kW × h/t	600	1100	900
Contamination	\	\	\
Economic benefit yuan/t	>1000	<0	<0

## Data Availability

No new data were created in this report.

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
