# Peer review of "Research Progress of Elastomer Materials and Application of Elastomers in Drilling Fluid"

_polymers, 2023, doi:10.3390/polym15040918_

Round 1

Reviewer 1 Report

The article provides a good overview of Elastomer Materials. It shows the possibility of their application in drilling. A fair amount of research has been presented. Which shows the high level of the authors.

A small note - check the title of table 4 ("Elastic modulus and yield stress of PUSS-66 after self-repair at different temperatures" or "Elastic modulus and yield stress of PUSS-6 after self-repair at different temperatures")

Author Response

请参阅附件。

Reviewer 2 Report

It is my pleasure to review a valuable review article on the application of polymeric elastomers in drilling fluid. As a review article, the organization of the elements of the article has been done correctly, and since the article has been submitted in the Journal of Polymers, adding part 3 titled "Preparation of elastomer powders" can be a good idea to encourage the readers of the article. Some points are suggested to improve the article.

1. English language and style are fine/minor spell check required.

2. The following articles are recommended for content enrichment:

"A hybrid nanocomposite of poly (styrene-methyl methacrylate-acrylic acid)/clay as a novel rheology-improvement additive for drilling fluids." Journal of Polymer Research 26 (2019): 1-14.

"Increasing the working life and performance improvements of down whole mud motors using nanocomposite elastomer." Journal of Petroleum Research 29.1-98 (2019): 84-94.

Reviewer 3 Report

In this study, the authors tried to review the current progress of the application of elastomer materials in the drilling operation. The presented work is significant for the readers and researchers in the study area. Before it can be accepted, the following comments must be considered in the revision: 

1. It is expected to have an extensive literature review followed by an in-depth and critical analysis of the state of the art, and identify challenges for future research.

2. In page 12, the authors wrote a section on the graphene/elastomer nanocomposites. It will be good if they can add the mechanism of graphene oxide with the elastomer. How the performance of the elastomer can be improved by nanos?
